# Toll like Receptor signalling by *Prevotella histicola* activates alternative NF-κB signalling in Cystic Fibrosis bronchial epithelial cells compared to *P. aeruginosa*

**Anne Bertelsen**[1,2], **Stuart J. Elborn**[1,3], **Bettina C. Schock**[1]*

**1** Wellcome-Wolfson Institute for Experimental Medicine, Queens University Belfast, Belfast, United Kingdom, **2** Department of Medicine, University of Cambridge, Addenbrookes Hospital, Cambridge, United Kingdom, **3** Imperial College London, London, United Kingdom

* b.schock@qub.ac.uk

**Data Availability Statement:** All relevant data are within the manuscript and its Supporting Information files.

## Abstract

Cystic Fibrosis (CF), caused by mutations affecting the *CFTR* gene, is characterised by viscid secretions in multiple organ systems. CF airways contain thick mucus, creating a gradient of hypoxia, which promotes the establishment of polymicrobial infection. Such inflammation predisposes to further infection, a self-perpetuating cycle in mediated by NF-κB. Anaerobic Gram-negative *Prevotella* spp. are found in sputum from healthy volunteers and CF patients and in CF lungs correlate with reduced levels of inflammation. *Prevotella histicola (P. histicola)* can suppress murine lung inflammation, however, no studies have examined the role of *P. histicola* in modulating infection and inflammation in the CF airways. We investigated innate immune signalling and NF-kB activation in CF epithelial cells CFBE41o- in response to clinical stains of *P. histicola* and *Pseudomonas aeruginosa* (*P. aeruginosa)*. Toll-Like Receptor (TLR) expressing HEK-293 cells and siRNA assays for TLRs and IKKα were used to confirm signalling pathways. We show that *P. histicola* infection activated the alternative NF-kB signalling pathway in CF bronchial epithelial cells inducing HIF-1α protein. TLR5 signalling was responsible for the induction of the alternative NF-kB pathway through phosphorylation of IKKα. The induction of transcription factor HIF-1α was inversely associated with the induction of the alternative NF-kB pathway and knockdown of IKKα partially restored canonical NF-kB activation in response to *P. histicola*. This study demonstrates that different bacterial species in the respiratory microbiome can contribute differently to inflammation, either by activating inflammatory cascades (*P. aeruginosa*) or by muting the inflammatory response by modulating similar or related pathways (*P. histicola*). Further work is required to assess the complex interactions of the lung microbiome in response to mixed bacterial infections and their effects in people with CF.

**Funding:** AB received a PhD studentship from the Department for Employment and Learning (DEL), Northern Ireland, UK (https://www.nidirect.gov.uk/articles/department-economy-studentships). BCS was supported by a grant from Northern Ireland Chest Heart and Stroke (NICHS, 2014_15) (https://www.nicva.org/organisation/ni-chest-heart-stroke) The funders had no role in study design, data collection and analysis, decision to publish, or preparation of the manuscript.

**Competing interests:** The authors have declared that no competing interests exist.

## Introduction

Cystic Fibrosis (CF), is an autosomal recessive life-limiting disease, characterised by viscid secretions in multiple organ systems due to mutations affecting the *CFTR* gene, which codes for a cAMP-regulated chloride channel, found on epithelial surfaces, including the airways, pancreas, and intestine [1]. To date over 2000 mutations have been described, with a phenylalanine deletion (F508del) being the most common disease-causing mutation (present in ~90% of PWCF). The main causes of mortality and morbidity in CF are lung damage and progressive lung function decline and, attributed in part to the dehydrated mucus and bacterial infection with Gram-negative pathogens which cause a sustained inflammatory response in the CF lung [2]. The contribution of bacterial species such as *Pseudomonas aeruginosa (P. aeruginosa), Burkholderia cepacia* complex, *Haemophilus influenza* and *Staphylococcus aureus (S. aureus)* to inflammation in the CF lung have been extensively studied [3–5], however, a broader picture of the microenvironment in the CF lung has recently become apparent with advances in high fidelity next generation sequencing (NGS) [1]. This method has identified a diverse microbiome in the lungs of people with CF, including the presence of the Gram-negative anaerobic genus *Prevotella* spp. [6]. Members of this genus are also found in high abundance in the upper airway of healthy people [7], but its role in the airways is poorly characterised. In people with CF the presence of *Prevotella* spp. has a positive correlation with higher lung function and reduced C-reactive protein (CRP) [8]. Furthermore, we recently showed that a strain of *P. nigrescens* isolated from a person with CF induced a lower pro-inflammatory cytokine expression in CF bronchial epithelial cells than *P. aeruginosa*, suggesting that the presence of certain *Prevotella* species in CF lungs may lower the inflammatory response and could therefore be beneficial to the host [9].

Interaction of bacterial pathogen the host cells (the airway epithelia cells) results in the activation of Toll-Like Receptors (TLRs) such as TLR2 (recognising mainly Gram-positive bacteria) and TLR4 (recognising mainly Gram-negative bacteria). In airway epithelial cells, such TLR-ligand binding activates NF-κB signalling leading to the induction of the innate inflammatory response.

NF-κB activation occurs via two major signalling pathways, the canonical and the non-canonical NF-κB signalling pathways. Common to both is the activation of the IκB kinase (IKK) complex (IKKα/IKKβ) [10]. In the canonical signaling pathway, binding of ligands to cell surface receptors such as Toll-like receptors (TLRs) lead to the recruitment of adaptors such as TRAF to phosphorylate IKK. Subsequent phosphorylation and degradation of IκB activates and translocates NF-κB dimers comprising RelA(p65), c-Rel, RelB and p50 to induce target gene expression [11]. Activation of the non-canonical pathway, involves the two IKKα subunits containing IKK complex, but not NEMO. Ligand-induced activation triggers NF-κB inducing kinase (NIK) to phosphorylate and activate the IKKα complex. In turn, IKKα phosphorylates p100 leading to the p52/RelB active heterodimer, important e.g. for the generation of B and T lymphocytes. Importantly, to date, only a small number of stimuli are known to activate the non-canonical pathway [12].

In CF airway diseases, the canonical NF-κB signalling pathway, consisting of the p65 and p50 subunits, and its role in inflammation in CF has been extensively investigated [9, 13, 14], however, the role of the alternative NF-κB signalling pathway, involving RelB and p52 subunits, has not. Activation of the alternative pathway is known to occur in response to a small set of agonists including lipopolysaccharide (LPS), CD40Ligand and lymphotoxin-alpha (LT-$\alpha_1$). This pathway has been shown to be active in epithelial cells and has recently been implicated in pathway analysis of inflammation in people with CF [15–17]. Inhibitor of kappaB kinase α (IKKα) plays a role in resolving NF-κB-driven inflammation in TLR expressing cells

by facilitating turnover of p65 and cRel, removing them from the nucleus [18]. Thereby IKKα activation was also shown to limit the inflammatory response during bacterial infection and inhibit canonical NF-κB activation [18].

Furthermore, mucus plugging and bacterial bio-film development results in the formation of gradients of hypoxia in the CF lung supporting induction of the transcription factor hypoxia induced factor-1α (HIF-1α). The relationship between induction of HIF-1α and NF-κB signalling indicates that the induction of HIF-1α may exert a regulatory effect on p65-driven inflammation [19]. However, the effect of HIF-1α on the alternative NF-κB pathway and possibly the resolution of inflammation is not clearly understood [20].

No studies have examined if infection with *P. histicola* results in the activation of TLR signalling pathways in CF bronchial epithelial cells and if TLR signalling can directly activate the alternative NF-κB signalling pathway in CF bronchial epithelial cells. As bacterial infection results in NF-κB signalling and induction of inflammatory cytokines, we hypothesised that *P. histicola* induces TLR signalling contributing to the activation of NF-κB signalling. As the presence of *Prevotella* spp. in CF lungs correlates with better lung function [8], we further hypothesised *P. histicola* may activate the alternative NF-κB driven response, which may inhibit canonical NF-κB activation [21]. These studies demonstrate that *P. histicola* activates the alternative NF-κB signalling pathway and HIF-1α in CF bronchial epithelial cells through TLR5 signalling.

## Materials and methods

### Bacterial culture

The bacterial isolates used in this study were all obtained from patients attending the adult CF clinic at Belfast City Hospital. The isolates were derived from two different patients enrolled in a multicentre study (Office for Research Ethics Committees Northern Ireland (OREC) 10/NIR01/41; Integrated Research Approval System (IRAS) Project no. 41579) as previously described [17].

The clinical isolate of *P. histicola* B011L was cultured under anaerobic conditions for 72 hours on Columbia Blood Agar (CBA, Fannin LIP) using a Don Whitley anaerobic cabinet (Don Whitley A35 workstation) as described [6]. This lawn of colonies was used to inoculate 10mL of anaerobic basal broth (Oxoid)TO OD.0.1 and this was allowed to grow to mid log phase (approximately 18 hours). This culture was used for infection experiments. *P. histicola* was identified by 16S rRNA sequencing, PGFE and RAPD analysis as described [9, 22].

*P. aeruginosa* (clinical isolate B021, identified using 16S rRNA screening [22]) was grown under aerobic conditions on Columbia blood agar (CBA) (37˚C, 5% $CO_2$, 95% mixed gas) over night. This culture was then utilised to inoculate a 10 mL culture of Lysogeny Broth (LB broth), The start OD was 0.05. This broth culture was incubated for up to two hours at 200 rpm, 37˚C until mid-log phase under aerobic conditions (approximately 3–4 hours) prior to being used for further cell infections under anaerobic conditions.

The minimum amount of bacteria required to provoke a significant response from CFBE41o- cells (0-4h, anaerobic conditions) was determined by screening of 3 different *P. aeruginosa* isolates as described [9]. Growth curves of *P. histicola* and *P. aeruginosa* under anaerobic conditions revealed no differences in the growth rates between the two species (S1 Fig).

### Cell culture

The F508del homozygote cystic fibrosis cell line (CFBE41o-) was maintained in antibiotic free minimum essential media (MEM, Gibco), supplemented with 10% heat inactivated foetal bovine serum (FBS, Gibco) and 5% L-Glutamine (Gibco) under standard cell culture

conditions (37˚C, 5% $CO_2$, 95% mixed gas). All tissue culture flasks and plates were pre-coated with a 1% PurCol type 1 collagen solution (Nutacon) and passaged as described previously [23]. HEK-293-TLR2, HEK-293-TLR4, HE-293-TLR5 and HEK-293-TLR null cells were maintained as per manufacturer's instructions (InvivoGen).

## Infection assays

CFBE41o- cells were infected with *P. histicola* or *P. aeruginosa* at an MOI (Multiplicity of Infection) of 100 for 4 hours. Both bacteria were grown to mid log phase as described previously [9]. An MOI of 100 was defined by plating all inocula on CBA agar and enumerating viable counts the following morning. Liquid cultures were used to inoculate cells for infection experiments and cells were incubated for up to 4 hours under anaerobic conditions as described above. Cells were incubated under anaerobic conditions for the duration of the experiments as described in supplementary data (S1 Text).

## Cell viability assays

To confirm that experimental conditions would not negatively affect cell viability, lactate dehydrogenase release (LDH, Abcam), mitochondrial respiration (MTT) and trypan blue exclusion were assessed after exposure to hypoxia and bacteria.

Briefly, bell death was assessed by measuring LDH release from infected cells and non-infected control cells. 10μL of supernatant was used for each assay as per manufacturer's instructions (Abcam, ab69693). Analyses of mitochondrial activation (measured by MTT (3- [4, 5-dimethyl thiazol-2yl]– 2, 5 diphenyl tetrazolium bromide) conversion to purple formazan (absorbance λ = 570nm)) served as a surrogate for cell viability. *Trypan Blue* (Sigma) was used in the dye *exclusion assay.* After incubation and loading onto a Neubauer haemocytometer, cells which appeared blue under the microscope were determined as 'dead' and cells appearing white were counted as 'live cells'. Further details of these assays can be found in the supplement (S2A–S2C Fig).

## TLR reporter assays

HEK-293-TLR2, HEK-293-TLR4 and HEK-293-TLR5 cells were purchased from InvivoGen and cultured and transfected as per manufacturer's instructions. Briefly, HEK-293 cells were maintained in high Glucose DMEM with 10% FBS, L-Glutamine and Pen/Strep. 100μg Blasticidin was added to cells after the second passage and cells were maintained in the media thereafter. Cells were transiently transfected with an NF-κB containing reporter construct plasmid (LyoVec and pNifty-Luc™, InvivoGen) and cells were incubated for 24 hours under standard tissue culture conditions to recover from the transfection. Bacterial infection was carried out as described [9] and cells were incubated under anaerobic conditions for the duration of the experiments.

## Cytoplasmic and nuclear fraction extraction for DNA binding ELISA

CFBE41o- cells were infected with either *P. aeruginosa* or *P. histicola* as described. DNA-binding ELISAs (enzyme-linked immunosorbent assays) were commercially sourced. **(i)** IKKα: Solid phase ELISA (Cell signal, CST, PathScan Phospho-IKKα). IKKα was quantified from all samples at 0–240 minutes as per manufacturer's instructions. **(ii)** IκBα (Inhibitor of kappa-B alpha): Sandwich ELISA to quantify phosphorylated IκBα in infected and non-infected cells at 0–120 minutes after infection (Function ELISA IκBα, S32/S36, Active Motif). **(iii)** NF-κB: DNA-binding ELISA to quantify active p65, p50, RelB and p52 subunits in infected and non-

infected cells (Trans/AM NFκB, Activ Motif). Nuclear cell fractions were generated using the NE-Per™ nuclear and cytoplasmic extraction reagents (Thermo Fisher Scientific).

### RT-PCR for TLR and cytokine gene expression

Total RNA was extracted from infected and un-infected control cell populations using a combination of TRIzol™ extraction reagent and column extraction as per manufacturer's instructions (Ambion/Thermo Fisher Scientific). Deoxyribonuclease (DNAse) digestion was carried out as per manufacturer's instructions using DNAse 1 (Thermo Fisher Scientific) for each sample. Reverse transcription was carried out using 0.5μg total RNA for each reaction (Superscript III reverse transcription kit, Thermo Fisher Scientific).

Quantitative reverse transcription PCR (qRT-PCR) analysis was carried out on an MX300-P (Agilent technologies) using gene specific primers (QuantiTect®, Qiagen) and SyBr Green PCR master mix (Thermo Fisher Scientific) as per manufacturer's instructions. The endogenously expressed beta-2-microglobulin (B2M) was used as a house-keeping gene in all experiments. Relative gene expression was calculated as $2^\wedge - \Delta\Delta CT$.

### siRNA knockdown assays

Silencer RNA (siRNA) targeted against TLR2, TLR5, scrambled siRNA and siRNA targeted against IKKα (Silencer select, Thermo Fisher Scientific). Lipofectamine 2000 was utilised and reverse transfection was used in all siRNA assays (Thermo Fisher Scientific). Target gene and protein expression were assessed at 24–48 hours post transfection. Lipofectamine did not induce any significant increase in LDH release (S3 Fig). SiRNA and Lipofectamine concentrations were optimised to induce a significant reduction in target gene expression (S4A Fig) and subsequent cytokine release (S4B Fig).

### Statistical analysis

All statistical analyses were performed using GraphPad Prism (v5.0; GraphPad, San Diego, CA). All data are expressed as mean +/- standard deviation (SD). Parametric tests including Student's t-test were employed to assess the null hypothesis. Where more than two groups were compared either one- or two-way ANOVA was utilised to assess differences between the mean or differences between matched samples with a post-hoc Bonferroni analysis performed for all tests utilising one-way ANOVA.

Further information to materials & methods can be found in the supplement (S1 Text).

## Results

### Inflammatory pathway analyses for *P. histicola*

To test the hypothesis that infection with *P. histicola* in CF airway epithelial cells may contribute to anti-inflammatory signalling through activation of the alternative NF-κB pathway we compared NF-κB signalling in CFBE41o- cells infected with *P. histicola* and *P. aeruginosa*.

Only *P. aeruginosa* infection increased phosphorylated IκBα (at 10 min vs 0 min) compared to non-infected control cells (p<0.01), which returned to baseline at 30 min post infection (Fig 1A). Infection with *P. aeruginosa* also induced significantly higher levels of nuclear p65-DNA binding (2 and 4 hours, p<0.001 and p<0.01, Fig 1B) and significantly higher nuclear p50-DNA binding in response to infection (at 4 hours, p<0.001, Fig 1C), but *P. aeruginosa* did not change nuclear RelB- or p52-DNA binding (Fig 1D and 1E). In contrast, CFBE41o- cells infected with *P. histicola* did not demonstrate elevated levels of phosphorylated IκBα at any time point and there was no change in nuclear p50 or p65 in response to infection

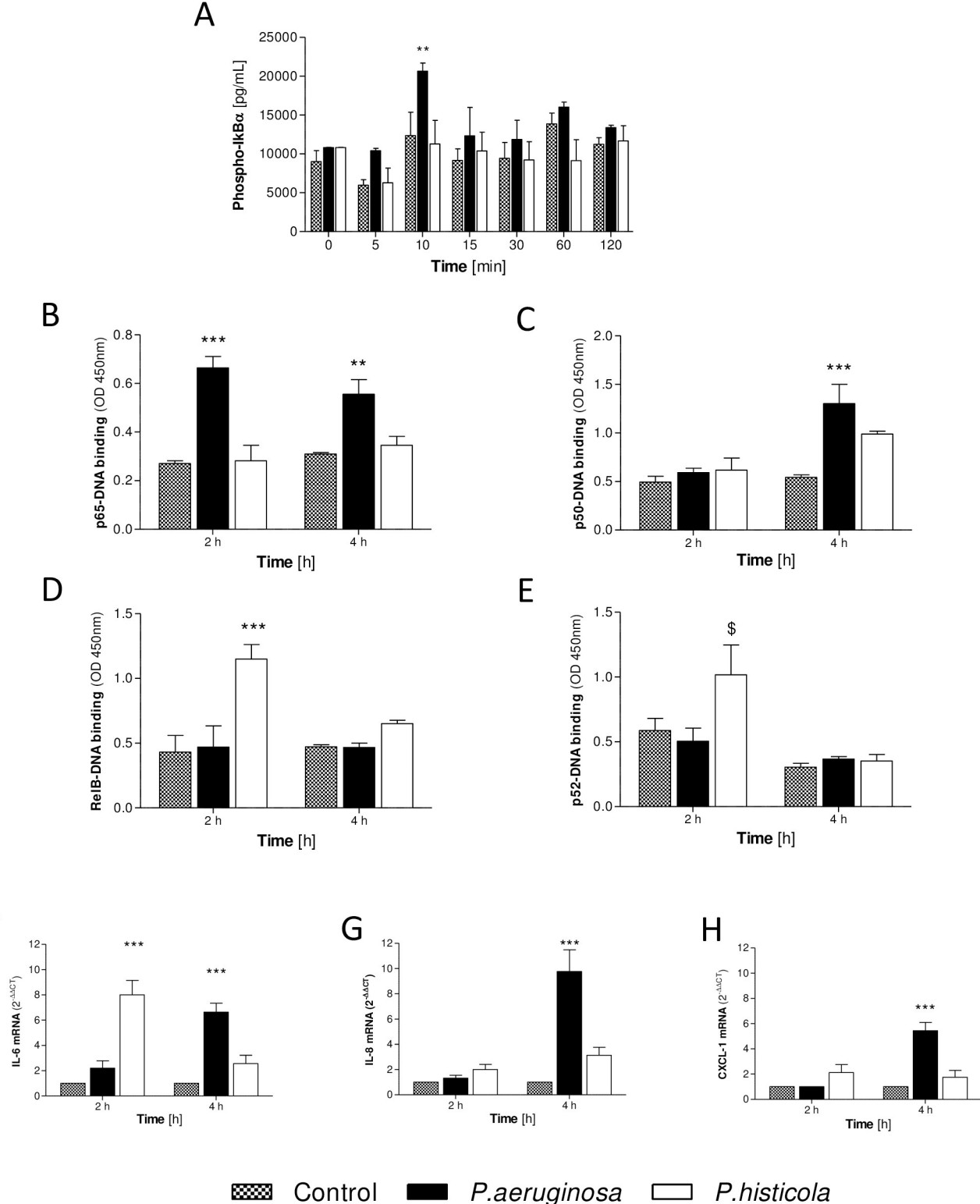

**Fig 1. *P. histicola*–Inflammatory pathway analyses.** CFBE41o- cells were infected with *P. aeruginosa* or *P. histicola* (MOI 100 for up to 4 hours, anaerobic conditions). **A**: Quantification of phosphorylated IκBα (pg/mL) in CFBE41o- cells infected with *P. aeruginosa* or *P. histicola* 0–60 minutes post infection. The DNA-binding abilities of p65, RelB and p52 were determined 2- and 4-hours post infection. **B**: DNA binding of p65 in infected and non-infected CFBE41o- cells. **C**: DNA binding of p50 in response to infection in CFBE41o- cells. **D**: RelB-DNA-binding in response to infection in CFBE41o- cells. **E**: DNA-binding of p52 in infected CFBE41o- cells. Quantitative RT-PCR using SyBr Green was carried out to determine mRNA

expression Beta-2-microglobulin was used as the endogenously expressed house-keeping gene for all experiments and $2^{-\Delta\Delta CT}$ was calculated for **F**: IL-6 mRNA, **G**: IL-8 mRNA and **H**: CXCL-1. 2-Way ANOVA with Bonferroni post-test, n = 3–5, *p<0.05, **p<0.01 and ***p<0.001 compared to non-infected CFBE41o-.

(Fig 1A–1C). However, significant levels of nuclear RelB were observed in response to infection with *P. histicola* (2 hours, p<0.001) with a reduction at 4 hours post infection compared to baseline controls (Fig 1D). Additionally, nuclear p52-DNA binding was increased 2 hours post infection with *P. histicola* with a significant reduction observed by 4 hours post infection (p<0.05, Fig 1E).

As we did not observe nuclear p65-DNA binding in response to infection with *P. histicola*, we hypothesised that infection with *P. histicola* would not result in transcription of NF-κB driven inflammatory cytokines. While *P. aeruginosa* infection resulted in significant induction of gene expression for interleukin (IL-)6, IL-8 and CXCL-1 in CFBE41o- cells (4 hours, Fig 1F–1H), there was no IL-8 or CXCL-1 gene induction in response to infection with *P. histicola* at any time point (Fig 1G and 1H). A significant increase in IL-6 gene expression was observed in CFBE41o- cells at 2 hours in response to infection with *P. histicola* (p<0.001, Fig 1F).

Taken together, these data indicate that *P. histicola* infection in CFBE41o- cells results in activation of the alternative NF-κB signalling pathway with no activation of the classic NF-κB pathway.

## *P. histicola* infection and TLR activation

*P. aeruginosa* signalling via TLR4 and TLR5 is well-established [24, 25], but little is known about *Prevotella*-induced TLR signalling. Therefore, to establish events upstream of NF-κB signalling utilised by *P. histicola*, we first investigated TLR receptor activation.

HEK-293-TLR4 cells demonstrated TLR4 signalling in response to infection with *P. aeruginosa* only (1–4 hours post infection, p<0.001), but no TLR4 signalling was observed after infection with *P. histicola* (Fig 2A). HEK-293-TLR2 cells showed TLR2 signalling 2 and 4 hours after *P. histicola* infection (both p<0.001, Fig 2B). *P. aeruginosa* infection lead to TLR2 signalling 4 hours after infection only (p<0.01, Fig 2B).

Furthermore, using HEK-293-TLR5 cells we observed TLR5 signalling 2 and 4 hours post infection with *P. aeruginosa* (p<0.01 and p<0.001) and also after *P. histicola* infection (p<0.001 and p0.01) with a significant reduction at 4 hours compared to 2 hours (p<0.01, Fig 2C).

To confirm that TLR2 and TLR5 signalling following *P. histicola* infection would lead to NF-κB activation, CFBE41o- cells were transiently transfected with siRNA targeted against TLR2 or TLR5 and infected with *P. histicola*. In response to infection we observed reduced levels of RelB DNA-binding in cells transfected with siRNA targeted against TLR5 (p<0.001, Fig 2D), but not when cells were transfected with siRNA targeted against TLR2 (Fig 3D). We also did not find any significant reduction in RelB DNA-binding in response to flagellin stimulation in these cells. However, flagellin alone significantly induces p65-DNA binding in CFBE41 cells (p<0.001, Fig 2E), but has no effect on RelB-DNA binding (Fig 2D). As a control, *P. aeruginosa* infection in TLR2 and TLR5 transfected cells did not lead to any significant changes in nuclear p65-DNA binding (Fig 2E).

Overall, these results show that *P. histicola* engages with CF epithelial cells via TLR5 and does not engage TLR2 signalling.

## IKKα in *P. histicola* infection

A role for IKKα in the regulation of canonical NF-κB-driven inflammation in macrophages has been described previously [18], but a role in the alternative NF-κB signalling is not known.

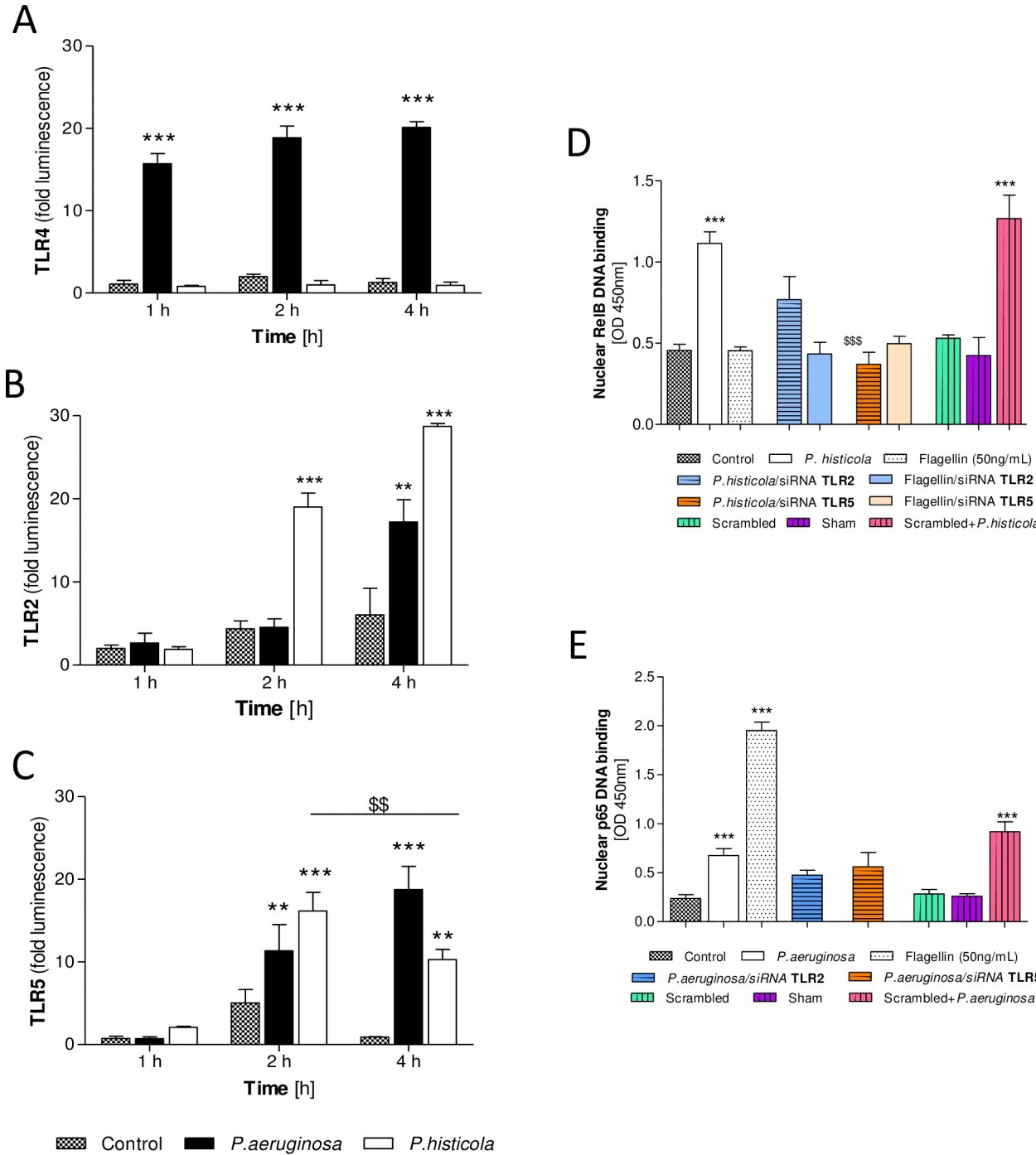

**Fig 2. *P. histicola*–TLR pathway signalling analyses.** Luminescence release from HEK-293-TLR2, HEK-293-TLR4, HEK-293-TLR5 or HEK-293-TLRnull cells was determined at up to 4 hours post infection normalised to the results observed from HEK-293-TLRnull cells. **A**: TLR4 signalling, **B**: TLR2 signalling; **C**: TLR5 signalling. 2-Way ANOVA with Bonferroni post-test, n = 3–5, ** p < 0.01 and *** p < 0.001 compared to non-infected CFBE41o-. CFBE41o- cells (infected with *P. aeruginosa*, *flagellin* or *P. histicola* under anaerobic conditions (MOI 100 for up to 4 hours) were transiently transfected with siRNA targeted against TLR2, TLR5, a scrambled transfection or a sham transfection. Nuclear protein was extracted 2 hours post infection and RelB-DNA binding was determined in transfected and non-transfected cells. **D**: Nuclear RelB-DNA binding in TLR2 and TLR5 siRNA transfected CFBE41o- cells. **E**: Nuclear p65-DNA binding in TLR2 and TLR5 siRNA transfected CFBE41o- cells. 1-Way ANOVA with Bonferroni post-test, n = 3–9, *** p < 0.001 compared to non-infected CFBE41o-; $$$ p < 0.001 compared to *P. histicola*.

To investigate the role of IKKα in NF-κB signalling, we determined phosphorylated IKKα in *P. histicola* infection and in CFBE41o- cells with transient IKKα knock down. CFBE41o- cells rapidly elevated phosphorylated IKKα in response to infection with *P. histicola* (5–60 min, p<0.05, p<0.001, Fig 3A), while infection with *P. aeruginosa* resulted in phosphorylation of IKKα at 60 minutes post infection only (p<0.001, Fig 3A). siRNA knockdown of IKKα resulted in a significant reduction in the DNA-binding abilities of nuclear RelB in response to infection with *P. histicola* (p<0.001, Fig 3B). As expected, *P. aeruginosa* infection significantly induced p65-DNA binding (p<0.001, Fig 3C), but *P. histicola* infection did not (scrambled IKKα siRNA and *P. histicola*, Fig 3C). siRNA IKKα in *P. histicola* infected CFBE41o- had no inducing effect on p65-DNA binding in CFBE41o-, and p65-DNA levels remained significantly lower compared to those in *P. aeruginosa* infected cells (p<0.001, Fig 3C). Our results confirm a role for *P. histicola*-induced IKKα in alternative NF-κB signalling.

## HIF-1α in *P. histicola* infection

HIF-1α, induced in response to anaerobic conditions, is thought to have a regulatory effect on NF-κB [26]. Therefore, we assessed if HIF-1α influenced NF-κB activation in response to infection with *P. histicola*. CFBE41o- cells had increased levels of HIF-1α protein in response to infection with *P. histicola* (1 and 4 hours, both p<0.01, Fig 4A). When CFBE41o- cells were transfected with siRNA targeting TLR2, intracellular HIF-1α levels remained elevated. However, in cells transfected with siRNA targeting TLR5, HiF-1α was significantly reduced in response to infection with *P. histicola* (30 min, p<0.001, Fig 4B) suggesting that *P. histicola* induced TLR5 signalling was mediated by HIF-1α.

## Discussion

In CF airways, infections with bacterial species such as *P. aeruginosa*, *Burkholderia cepacia* complex, *Haemophilus influenza* and *S. aureus* are well investigated [3–5]. However, advances in NGS have identified the presence of *Prevotella* spp. in the lungs of people with CF [6]. Moreover, no studies have examined the ability of TLR signalling to activate the alternative NF-κB signalling pathway or the role of transcription factor HIF-1α in this CF bronchial epithelial cells response [27]. Therefore, this study was set out to investigate the contribution of *Prevotella* spp. (*P. histicola*) in TLR and NF-κB signalling to CF airways inflammation.

The relative abundance of anaerobic bacteria compared to aerobic bacteria in CF sputum is associated with a milder disease when compared to *Pseudomonas*-dominated patients. In this large multisite study involving clinical stable CF patients from the UK, Ireland and the US, Muhlebach *et al.* showed the culture of 18 anaerobic genera from 59% of sputum samples (95% aerobic). Significantly prevalent anaerobes were *Prevotella* species, followed by *Veillonella*, *Porphyromonas* and others. *Importantly, the* prevalence of anaerobes was positively associated with pancreatic sufficiency, better nutrition and better lung function [28]. Consistent with this, reduced bacterial diversity and increased levels of inflammation have also been reported in patients with CF [8, 29]. A lower abundance of aerobic and anaerobic bacteria reflecting microbiota disruption was also shown to be associated with disease progression (lower lung clearance index (LCI) and higher CRP in the CF lung [29]. Zemanick *et al.* investigated the presence of anaerobes in early CF exacerbations showing the highest relative abundance for Prevotella, Veillonella and Porphyromonas. Higher levels of sputum anaerobes were associated with less inflammation and higher lung function compared to the presence of Pseudomonas at exacerbation [8]. However, hile the weak inflammatory properties of commensal *Prevotella* spp. might help the airways' immune system tolerate colonisation [30], *P. histicola* has also been shown to exert anti-inflammatory properties in a murine model of rheumatoid

A

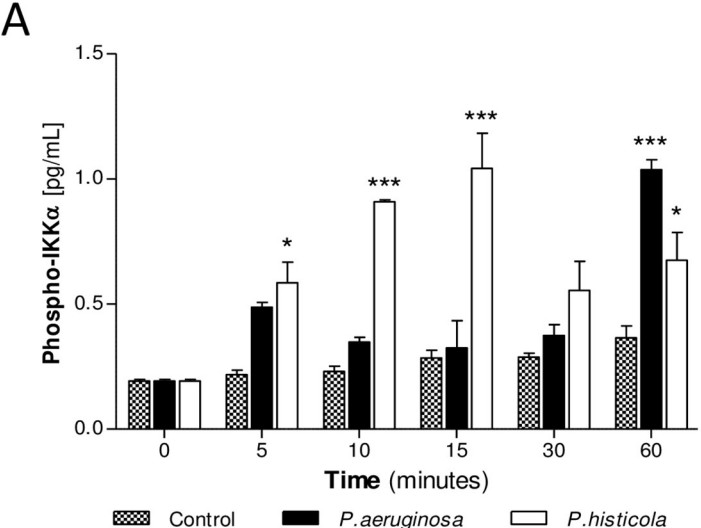

B

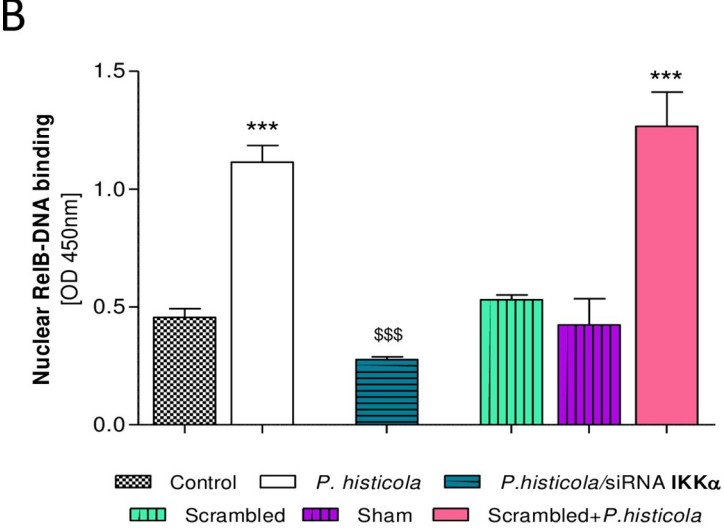

C

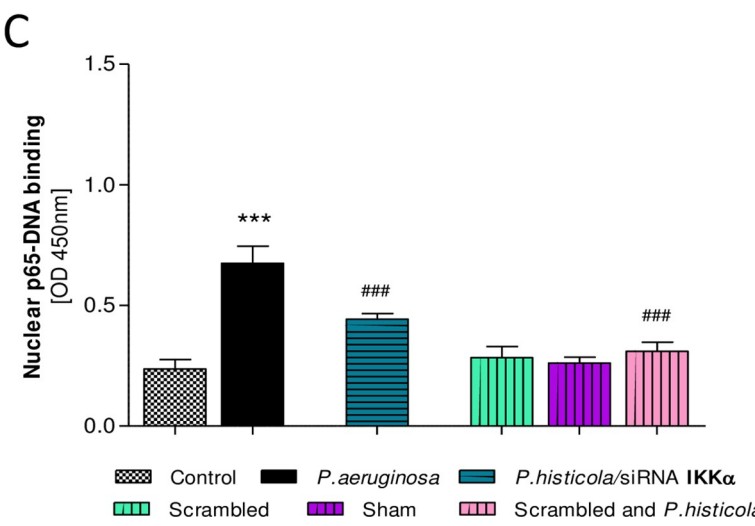

**Fig 3. IKKα in *P. histicola* infection.** Phosphorylated IKKα in infected and non-infected CFBE41o- cells (MOI 100, *P. aeruginosa* and *P. histicola)* determined at 0–60 minutes. CFBE41o- cells were also transiently transfected with siRNA targeted against IKKα. RelB and p65-DNA binding was determined in *P. histicola* infected cells (MOI 100, for 2 hours). **A**: Time course IKKα phosphorylation in CFBE41o- cells in response to infection. 2-Way ANOVA with Bonferroni post-test, n = 3–5, *p<0.05 and ***p<0.001 compared to non-infected CFBE41o-. **B**: Nuclear RelB-DNA binding in response to transfection with IKKα siRNA and infection with *P. histicola*. **C**: Nuclear p65-DNA binding in response to transfection with IKKα siRNA and infection with *P. histicola*. 1-Way ANOVA with Bonferroni post-test, n = 3–9, ***p<0.001 compared to non-infected CFBE41o-; $^{$$$}$p<0.001 compared to *P. histicola* (RelB) *or P. aeruginosa* (p65).

arthritis [31]. Our data confirm that *P. histicola* contributes to anti-inflammatory signalling, facilitated through activation of the alternative NF-κB pathway via RelB and p52, while *P. aeruginosa* signalling is through IκBα phosphorylation and p65/p50 activation.

Hypoxia can affect bacterial growth and Schaible *et al*. showed that *P. aeruginosa* growth was slower under hypoxic conditions than under normoxic conditions [32]. However, comparing *P. histicola* and *P. aeruginosa* under anaerobic conditions revealed no differences in growth rates between the two species (S1 Fig), suggesting that any differences described are a direct result of a different pathway activation.

Our subsequent investigation of TLR receptor activation demonstrated a role for *P. histicola* in TLR5-induced activation of the alternative NF-κB signalling in CF bronchial epithelial cells. Additionally, we demonstrate a role for IKKα and the induction of HIF-1α in regulating CF inflammation. TLR signalling and the resulting inflammation has been extensively studied in CF. TLRs (especially TLR2, 4 and 5) in the CF lung are one of the most common innate immune defences activated in response to infection leading to the activation of the classical NF-kB-activation and subsequent pro-inflammatory cytokine release [33–36].

Using TLR transfected HEK-293 reporter cells, we have shown that *P. histicola* infection induced TLR2 signalling (at 4 hours) and importantly a rapid induction in TLR5 signalling, which was significantly reduced by the end of the 4 hour infection period. Having established the TLR signalling route of *P. histicola* in CF bronchial epithelial cells, we further investigated the downstream signalling. Infection with *P. histicola* resulted in the induction of RelB and p52 protein but not p65 or p50 protein, clearly indicating the induction of the alternative NF-κB pathway. Alternative NF-κB signalling is involved in cell survival and proliferation and has been shown to have anti-inflammatory effects. Studies in RelB deficient mice demonstrated an anti-inflammatory role for RelB [37, 38]. In human lung fibroblasts, RelB overexpression reduced IL-1β-induced inflammation [39]. Furthermore, RelB has been linked to the development of tolerance in THP-1 cells [40] through repression of TNFα and Il-1β [41]. Mechanistically, RelB may regulate IkBα stability and may thereby limit the canonical NF-kB activation [42]. Others have suggested that RelB may interfere with NF-kB activity in the nucleus through protein–protein interactions with RelA to form inactive complexes leading to down-regulation of NF-κB target genes [43–45]. Alternative NF-κB signalling through anti-inflammatory RelB may therefore explain why in patients with CF *Prevotella* spp. in sputum have been associated with improved lung function and reduced inflammation in CF [8]. This is further supported by our findings that gene expression of the canonical NF-κB-dependent cytokines IL-8 and CXCL-1 were not observed at any time point in response to infection with *P. histicola*. Increased IL-6 gene expression was observed at 2 hours post infection with *P. histicola* with a reduction at 4 hours, indicating a possible role for the alternative NF-κB signalling pathway in the regulation of IL-6 gene expression, or/and a role for IKKα in the regulation of IL-6 gene expression [46].

Our data demonstrate signalling through TLR5 by the clinical *P. histicola* strain used. The siRNA experiments not only confirmed the activation of TLR5 signalling in CF bronchial

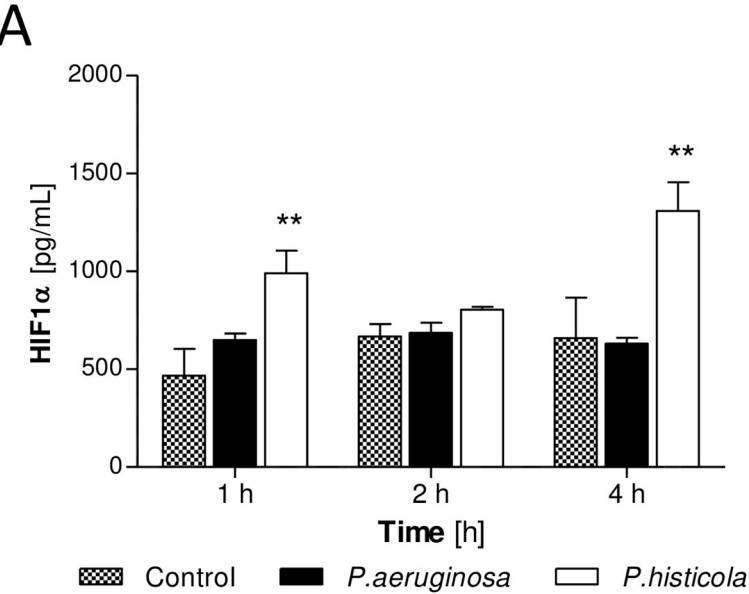

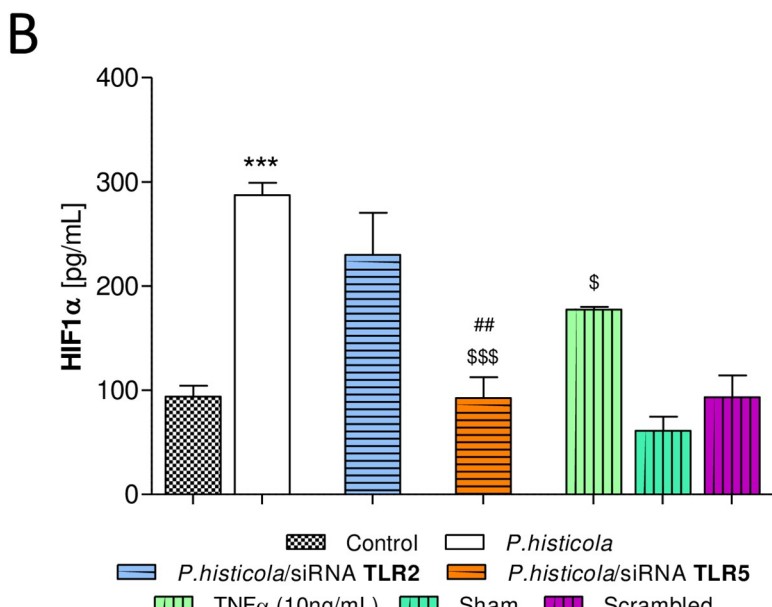

**Fig 4. HIF-1α in *P. histicola* infection. A**: Time course HIF-1α protein induction in CFBE41o- cells in response to infection. CFBE41o- cells were transfected with *P. aeruginosa* or *P. histicola* (MOI 100) and incubated under anaerobic conditions for up to 4 hours. Total HIF-1α protein was determined at 1, 2 and 4 hours post infection. 2-Way ANOVA with Bonferroni post-test, n = 3, **p<0.01 compared to non-infected CFBE41o-. **B**: CFBE41o- cells were transiently transfected with siRNA targeted against TLR2 and TLR5, infected with *P. histicola* (MOI 100) and incubated under anaerobic conditions for 30 minutes. Total HIF-1α protein was determined in transfected and non-transfected cells in response to infection. 1-Way ANOVA with Bonferroni post-test, n = 4, ***p<0.001 compared to non-infected CFBE41o; $p<0.05, $$$p<0.001 compared to *P. histicola*; ##<0.01 *P. histicola*/siRNA TLR2 vs TLR5.

epithelial cells in response to *P. histicola*, but also that the induction of RelB in *P. histicola* infection was a downstream result of TLR5 signalling. Using flagellin alone to activate TLR5, we clearly show that such activation leads to NF-κB(p65) activation and does not include activation of RelB. So far, flagellin is the main ligand identified for TLR5 [47]. In *P. aeruginosa*

flagellin signalling through TLR5 is well-established [48, 49] and induces the canonical pro-inflammatory NF-κB pathway [50–52]. In several Prevotella species none or only one flagella synthesis pathway gene has been identified [53]. Therefore, our data may suggest that *Prevotella spp.* could use other structures for attachment and motility and TLR5 engagement. In support of this, recent work in neurons and neuropathic pain strongly suggests TLR5 activation could also be facilitated by non-flagellin small molecules [54]. Further, *H. pylori* has been shown to activate TLR5 through CagL, a component of its type IV secretion system, which acts as a flagellin-independent TLR5 activator [55]. While our experiments show TLR5 signalling by *P. histicola*, we did not further investigate flagellin expression in the *P. histicola* strain used, nor did we investigate non-specific TLR5 activation in our experimental cell model. Furthermore, we used flagellin derived from Salmonella, which was highly pro-inflammatory, but we cannot exclude that other flagellins (e.g. from *P. aeruginosa*) may provoke a response of different intensity.

As we observed a strong phosphorylation of IKKα induced by *P. histicola*, we further sought to establish if such IKKα activation was involved in the activation of the alternative NF-κB signalling pathway and if this would have any regulatory effect on the canonical pathway. Knockdown of IKKα significantly reduced *P. histicola* induced activation of the alternative NF-κB signalling (RelB-DNA binding), but we saw an induction in p65 (see 'scrambled and *P. histicola*' in Fig 3C) indicating a role for IKKα in the regulation and suppression of p65.

In normoxia, HIF expression is regulated through hydroxylation by prolyl-4-hydroxylases (PHDs) and subsequent polyubiquitination promotes HIF degradation [56]. However, epithelial stretch has been shown to stabilise HIF1α (through inhibition of succinate dehydrogenase (SDH) [57]. During hypoxia, the activity of PHDs is suppressed, allowing HIF-1α to translocate into the nucleus where it binds to its dimerization partner HIF1B and enhances the transcription of HIF target genes [56]. HIF-1α stabilization during hypoxia is achieved through RSUME/SUMO, where RSUME (RWD-containing sumoylation enhancer) increases the conjugation of SUMO (small ubiquitination modifier) and stabilizing HIF-1α [26]. IKBα is similarly stabilized during hypoxia, inhibiting NF-κB activation and signalling, indicating a role for hypoxia induced HIF-1α in the regulation of NF-κB [26]. In support of this regulatory function of HIF-1α, pharmacological *in vivo* studies in acute lung injury (ALI) have shown that HIF-1α stabilization attenuates pulmonary oedema and lung inflammation [57]. In our study HIF-1α protein was induced in response to infection with *P. histicola* (in hypoxic conditions) mediated by TLR5, suggesting that TLR5 in addition to inducing inflammation may also have a regulatory role in the inflammatory response through HIF-1α. HIF-1α activation induces several cytokines such as TNFα, IL-1, IL-4, IL-6 and IL-12 [58]. After *P. histicola* infection, we show a significant upregulation of IL-6 expression, a cytokine known for both pro-inflammatory and anti-inflammatory actions [59].

Furthermore, RelB may exert a suppressive effect on HIF-1α as we observed an inverse relationship between IKKα induction (Fig 3A) and HIF-1α expression (Fig 4A). However, as we have analysed total HIF1α, further research will be required to substantiate this hypothesis.

Limitations to the experimental design include the inability of the CFBE41o- cells to remain viable and responsive to infection past four hours. In contrast, a study by Schaible *et al.* acclimatised airway epithelial cells to hypoxia by incubation in hypoxic condition for 24 hours, although for CFBE41o- cells no information of cell viability is given [32]. Using 3 different assays to determine cell viability (MTT formation, LDH release, trypan Blue exclusion assay), in our cell culture system CFBE41o- cells show a significant reduction in cell viability 6 hours after the introduction of hypoxia (S2A–S2C Fig), which made it necessary to perform experiments for 4 hours. Although this appears to be a short time point for cytokine release, using a similar time point, Veit *et al.* showed detectable cytokine levels in CFBE41o- cells [60].

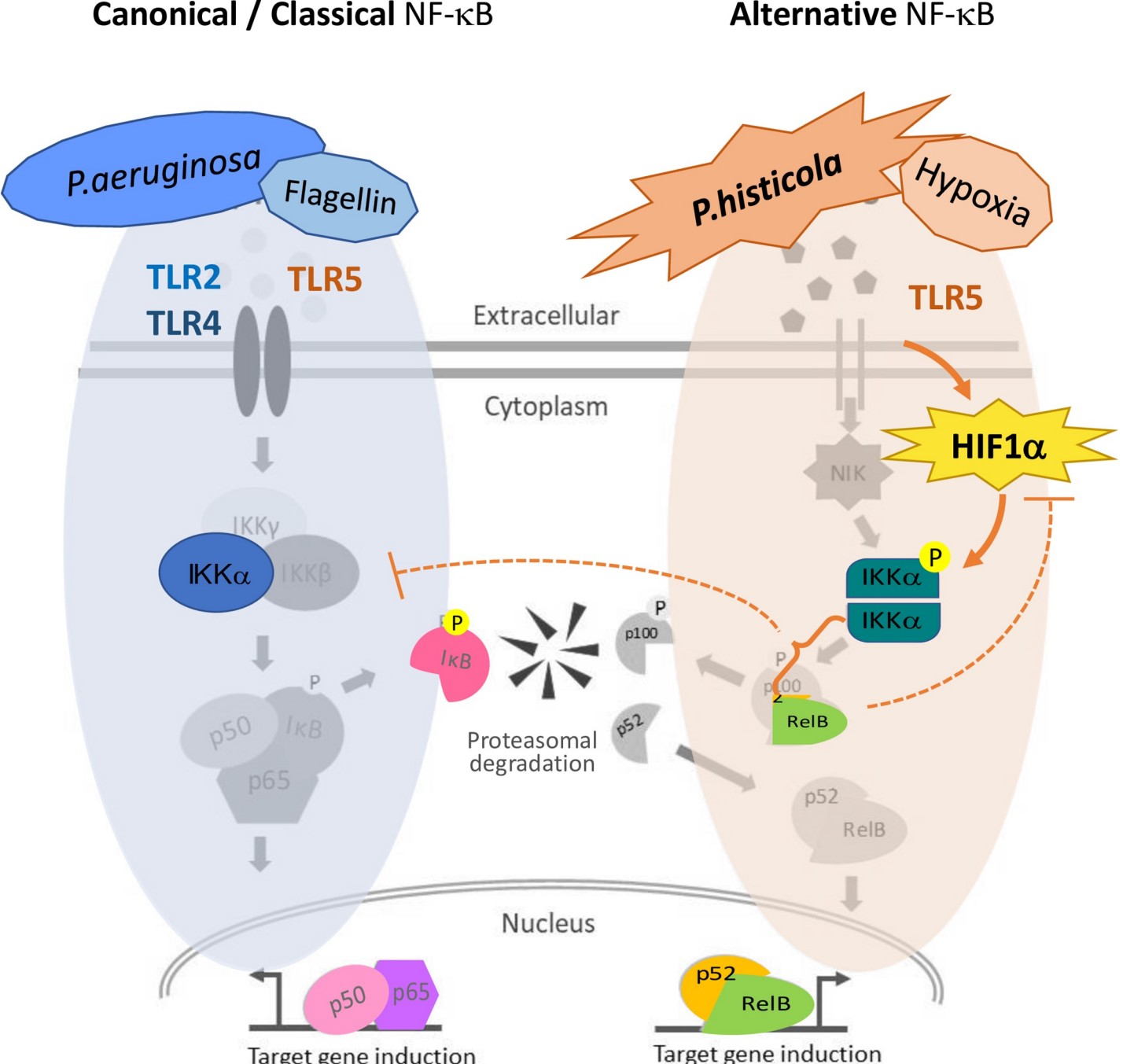

**Fig 5. *P. histicola* signalling–Schematic representation.** Canonical NF-κB signalling pathway (left) utilising p65/p50 and the alternative NF-κB signalling pathway (right) signalling through RelB. While *P. aeruginosa* and flagellin signal through TLR4 / TLR5-NF-κB (p65), *P. histicola* signals through TLR5 activating the alternative NF-kB (RelB) pathway. HIF1α mediates this signalling through activation of IKKα. However, RelB might introduce a negative feedback on HIF-1a and may also inhibit IKbα. Proteins determined within the study are highlighted in colour.

A further limitation of this study is the use of only one *P. histicola* clinical strain. We opted for the comparison of a clinical *P. histicola* strain with a clinical *P. aeruginosa* strain to reflect clinical conditions. Finally, this study was performed in one CF epithelial cell line (CFBE41o- cells) and, as we were interested in the different responses of CF relevant pathogens

(*P. histicola* versus *P. aeruginosa*), we did not perform the experiments in non-CF airway cells or CFTR corrected CF epithelial cells. TLR signalling was also investigated using human embryonic kidney (HEK) 293 reporter cells (InvivoGen), which are specially engineered to constitutively express a given functional pathogen recognition receptor gene and are not expected to express CFTR. Our results need to be confirmed using differentiated primary epithelial cells (CF and non-CF), particularly since mucus production would be expected to contribute to anaerobic conditions in the airway.

Despite these limitations, this study is the first to show the signalling pathway of *P. histicola* in CF bronchial epithelial cells leading to TLR5 activation resulting in the activation of the alternative NF-κB signalling pathway. Our work also raises the possibility of a flagellin-independent process, which needs further investigation. This is also the first study showing TLR5-RelB signalling for a clinical isolate of *P. histicola*. Furthermore, we have identified a role for IKKα in the regulation of inflammation in response to infection with *P. histicola* and this may explain the correlation between elevated levels of anaerobes in the microbiome and better lung function in CF patients [29]. *Prevotella* spp. in the gut microbiome display substantial genomic diversity between strains [61], but these studies have also identified beneficial and detrimental roles for these bacteria, demonstrating that each species may have dual roles in the microbiome and that further studies are required before we can fully appreciate the role of *Prevotella* spp. in disease and health [62–64].

## Conclusion

Anaerobic bacteria such as Gram-negative *Prevotella* spp. are frequently found in the airways from healthy volunteers and CF patients, but no studies have examined the role of *P. histicola* in modulating inflammatory pathways in CF airway. Our study demonstrates for the first time important differences in the activation of the NF-kB pathway in anaerobic CF lung inflammation between *P. histicola* and *P. aeruginosa*, and shows a relationship between the alternative NF-κB signalling pathway (utilised by *P. histicola*) and the HIF-1α signalling pathway (summarised in Fig 5). Overall, our work suggests that different species of bacteria present in the respiratory microbiome can contribute differently to inflammation in the CF lung, either by activating inflammatory cascades (e.g *P. aeruginosa*) or by muting the inflammatory response to infection by modulating similar or related pathways (as shown for *P. histicola*). Further work is required to assess the complex interactions of the lung microbiome in response to bacterial infections and their positive effects in people with CF.

## Supporting information

**S1 Text. Materials & methods.**
(DOCX)

**S1 Fig. Growth rate of *P. histicola* and *P. aeruginosa*.** Growth rate of *P. histicola* B011L and *P. aeruginosa* B021 under anaerobic conditions as described in supporting information above. All data n = 3–4 with 2way ANOVA with Bonferroni correction.
(TIF)

**S2 Fig. Cell viability under aerobic and anaerobic conditions. (A)** Mitochondrial activity of CFBE41o- cells (MTT assay); (**B**) % LDH release from CFBE41o- cells and (**C**) uptake of Trypan blue by CFBE41o- cells incubated under anaerobic and aerobic conditions as described in supporting information above. All data n = 4 with 2way ANOVA with Bonferroni correction, $^*p<0.05$, $^{**}p<0.01$.
(TIF)

**S3 Fig. Cell viability during transfection.** Lactate dehydrogenase (LDH) release from CFBE41o- cells after 0-96h of transfection with Lipofectamine 2000 (0.5–1.5 ng/ml) in the presence and absence of ALL-STARS Hs cell death siRNA (Qiagen) to assess transfection efficiency. All n = 3, *p<0.05 compared to control. 2-Way ANOVA with Bonferroni post-test, overall ANOVA given within the graph.
(TIF)

**S4 Fig. Transfection efficiency for target genes.** Efficiency for transfection was assessed by transfecting cells with TLR5 siRNA followed by the addition of either *P. histicola* (B011L), *P. aeruginosa* and flagellin and calculated relative to the induced response. **(A)** Significant reduction in TLR5 mRNA (55% and 60% in response to *P. histicola* and *P. aeruginosa* infection, respectively). **(B)** This was followed by a ≥70% reduction in IL-6 mRNA (48 hours post transfection).
(TIF)

## Acknowledgments

The authors thank all technical staff of QUB and the Belfast Trust for their help.

## Author Contributions

**Conceptualization:** Stuart J. Elborn.

**Data curation:** Anne Bertelsen, Stuart J. Elborn.

**Formal analysis:** Anne Bertelsen, Bettina C. Schock.

**Funding acquisition:** Stuart J. Elborn.

**Investigation:** Anne Bertelsen.

**Methodology:** Anne Bertelsen, Bettina C. Schock.

**Resources:** Stuart J. Elborn.

**Supervision:** Stuart J. Elborn, Bettina C. Schock.

**Validation:** Anne Bertelsen.

**Visualization:** Anne Bertelsen, Bettina C. Schock.

**Writing – original draft:** Anne Bertelsen.

**Writing – review & editing:** Stuart J. Elborn, Bettina C. Schock.

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
