## [Decision Letter · Decision Letter 0]

13 Aug 2020

PONE-D-20-19173

TLR5 signalling activates the alternative NF-κB signalling pathway in Cystic Fibrosis bronchial epithelial cells in response to infection with Prevotella histicola.

PLOS ONE

Dear Dr. Bettina C Schock,

Thank you for submitting your manuscript to PLOS ONE. After careful consideration, we feel that it has merit but does not fully meet PLOS ONE’s publication criteria as it currently stands. Therefore, we invite you to submit a revised version of the manuscript that addresses the points raised during the review process.

We look forward to receiving your revised manuscript.

Kind regards,

Abdelwahab Omri, Pharm B, Ph.D

Academic Editor

PLOS ONE

Journal Requirements:

2. Please amend either the title on the online submission form (via Edit Submission) or the title in the manuscript so that they are identical.

Reviewers' comments:

Reviewer's Responses to Questions

**Comments to the Author**

1. Is the manuscript technically sound, and do the data support the conclusions?

Reviewer #1: Yes

Reviewer #2: Yes

Reviewer #3: Yes

2. Has the statistical analysis been performed appropriately and rigorously? 

Reviewer #1: Yes

Reviewer #2: Yes

Reviewer #3: Yes

3. Have the authors made all data underlying the findings in their manuscript fully available?

Reviewer #1: Yes

Reviewer #2: Yes

Reviewer #3: Yes

4. Is the manuscript presented in an intelligible fashion and written in standard English?

Reviewer #1: Yes

Reviewer #2: Yes

Reviewer #3: Yes

5. Review Comments to the Author

Reviewer #1: Congrats on a good experimental design and valuable data! Please see my minor edits and suggestions throughout the manuscript.

In terms of the issue of higher HIF alpha expression level in healthy vs CF lungs (Review paper by Montgomery et al 2017), would be nice to indicate the microbiome diversity/dominant species

composition in relation to this report.

Reviewer #2: In this manuscript, Bertelson et al report comparative studies of the difference between the CF lung pathogen Pseudomonas aeruginosa and the airway commensal Prevotella histicola in inducing immune responses using cell culture models. In general I think the question is very interesting and the approaches used are reasonable. The figures are well put together and controls are well provided. Statistical analyses appear to be solid. The paper is generally well written.

Comments:

My most pressing concern is the paucity of methodological detail. Much more extensive detail should be added to methods sections. As an example, the “infection assays” section is a single sentence and is missing a great deal of pertinent information. As the manuscript currently stands, I doubt others would be able to recapitulate the results presented, based on the level of detail in the methods.

Growth rate comparison between P. aeruginosa and P. histicola is not standard. Cells should be inoculated at low enough density to monitor lag phase and tracked for long enough to capture log and stationary phases. Minimally, more than 1 doubling should be measured for a true growth rate calculation. It appears that doubling time here is ~4 hours, but at what part of the growth curve are we looking at? Furthermore, the methods here are lacking and the cited reference (Tunney et al 2007 AJRCCM) does not provide any detail germaine to the method at hand, perhaps it is the wrong reference? Regardless, the authors should provide full methodological detail here (and for other sections). For instance, how did the authors so precisely obtain the exact same number of cells reported for each species for the first time point? This result is important to bolster in order to support claims made by the authors (for instance on lines 360-361).

The authors use the term “infection” when referring to the experiments described in Fig 1. Do they mean that they are modeling infection of the airway by applying bacteria to cell culture? Or are they specifically referring to infection of their cells by bacteria becoming internalized? Commonly, for infection assays involving bacteria, there is a question of how many bacteria are internalized versus remain extracellular upon treatment. A useful method is to incubate bacteria with cultured cells, allow for internalization, then treat with antibiotic (e.g. gentamicin) to kill extracellular bacteria, and subsequent washing of cells. Assessment of bacterial internalization can then be performed by cfu plating or qPCR. Since the authors do not report doing this, presumably they do not know if bacteria are internalized or not, and if there are differences in the abundance of internalized Ph versus Pa bacteria. TLR5 signaling I believe does not require internalization, therefore perhaps what they are observing is detection of extracellular bacteria. If this is true, and/or if this is what the authors mean (as seems to be indicated by the graphic depiction in Figure 5), I recommend changing the term “infection” to something like “exposure”.

P.histicola and P.aeruginosa – there should be a space following “P.” throughout the text

Lines 90-91 should be cited. In general, more statements throughout the text that are presented as facts, should be supported by references to published literature. Another example is the first line of the Discussion lines 339-340, and lines 367-369.

Line 454, typo “aerugiosa”

Typo “.f” at the end of line 362.

Reviewer #3: The authors Bertelsen et al. have analyzed an alternate NF-kb signalling pathway activated by P. histicola in CF epithelial cells.

The initial analyses was performed in CFBE41o- cells and subsequently in HEK cells expressing various TLR ligands.

The findings are novel and indicate the anti-inflammatory effect of P. histicola.

Major comment:

1. The results should be validated in another CF cell line.

Minor comments:

1. The authors report the mRNA levels of IL-8, IL-6 and CXCL1 (Figure 1). They should also report the corresponding secreted protein levels.

2. The authors should proof read for typos.

3. Higher resolution images should be used for the figures.

6. PLOS authors have the option to publish the peer review history of their article (what does this mean?). If published, this will include your full peer review and any attached files.

Reviewer #1: **Yes: **Ali Azghani, Ph.D.

Reviewer #2: No

Reviewer #3: No

---

## [Author Response · Author response to Decision Letter 0]

16 Sep 2020

Manuscript Ref. No.: PONE-D-20-19173

Title: Toll like Receptor signalling by Prevotella histicola activates alternative NF-κB signalling in Cystic Fibrosis bronchial epithelial cells compared to P.aeruginosa

We thank the reviewers and the editor for their careful review of the manuscript and their overall very positive assessment. We have now addressed all issues raised and modified the manuscript in a revised version. 

Please find below our point-by-point response to the issues pointed out.

Journal Requirements:

This has been modified according to the PLOS ONE's style requirements.

Please amend either the title on the online submission form (via Edit Submission) or the title in the manuscript so that they are identical.

The title in the online submission has been changed to “Toll like Receptor signalling by Prevotella histicola activates alternative NF-κB signalling in Cystic Fibrosis bronchial epithelial cells compared to P. aeruginosa”.

Please include captions for your Supporting Information files at the end of your manuscript, and update any in-text citations to match accordingly. 

This has been modified according to the PLOS ONE's requirements.

Response to Reviewers' comments:

Reviewer #1:

Q1) In terms of the issue of higher HIF alpha expression level in healthy vs CF lungs (Review paper by Montgomery et al 2017), would be nice to indicate the microbiome diversity/dominant species composition in relation to this report.

A1) We have now added more detailed information about dominant anaerobe species into the manuscript (line 347-363). This now reads:

“The relative abundance of anaerobic bacteria compared to aerobic bacteria in CF sputum is associated with a milder disease when compared to Pseudomonas-dominated patients. In a large multisite study involving clinical stable CF patients from the UK, Ireland and the US, Muhlebach et al. showed the culture of 18 anaerobic genera from 59% of sputum samples (95% aerobic). Significantly prevalent anaerobes were Prevotella species, followed by Veillonella, Porphyromonas and others. Importantly, the prevalence of anaerobes was positively associated with pancreatic sufficiency, better nutrition and better lung function (Muhlebach, Hatch et al. 2018). Consistent with this, reduced bacterial diversity and increased levels of inflammation have also been reported in patients with CF (Zemanick, Harris et al. 2013, O'Neill, Bradley et al. 2015). A lower abundance of aerobic and anaerobic bacteria reflecting microbiota disruption was shown to be associated with disease progression (lower lung clearance index (LCI) and higher CRP in the CF lung (O'Neill, Bradley et al. 2015). Zemanick et al. investigated the presence of anaerobes in early CF exacerbations showing the highest relative abundance for Prevotella, Veillonella and Porphyromonas. Higher levels of sputum anaerobes were associated with less inflammation and higher lung function compared to the presence of Pseudomonas at exacerbation (Zemanick, Harris et al. 2013). However, while the weak inflammatory properties of commensal Prevotella spp. ….. “

Reviewer #2:

Q2) My most pressing concern is the paucity of methodological detail. Much more extensive detail should be added to methods sections. As an example, the "infection assays" section is a single sentence and is missing a great deal of pertinent information. As the manuscript currently stands, I doubt others would be able to recapitulate the results presented, based on the level of detail in the methods.

A2) We have now added more methodological detail into the manuscript. Further details can now also be found in the supplement (S1 Text, Materials & Methods).

The revised version now reads: 

“Materials and Methods

Bacterial culture

The bacterial isolates used in this study were all obtained from patients attending the adult CF clinic at Belfast City Hospital. The isolates were derived from two different patients enrolled in a multicentre study (Office for Research Ethics Committees Northern Ireland (OREC) 10/NIR01/41; Integrated Research Approval System (IRAS) Project no. 41579) as previously described (17).

The clinical isolate of P. histicola B011L was cultured under anaerobic conditions for 72 hours on Columbia Blood Agar (CBA, Fannin LIP) using a Don Whitley anaerobic cabinet (Don Whitley A35 workstation) as described (Tunney, Field et al. 2008). This lawn of colonies was used to inoculate 10mL of anaerobic basal broth (Oxoid)TO OD.0.1 and this was allowed to grow to mid log phase (approximately 18 hours). This culture was used for infection experiments. P. histicola was identified by 16S rRNA sequencing, PGFE and RAPD analysis as described (Gilpin, Nixon et al. 2017, Bertelsen, Elborn et al. 2019).

P. aeruginosa (clinical isolate B021, identified using 16S rRNA screening (Gilpin, Nixon et al. 2017)) was grown under aerobic conditions on Columbia blood agar (CBA) (37°C, 5% CO2, 95% mixed gas) over night. This culture was then utilised to inoculate a 10 mL culture of Lysogeny Broth (LB broth), The start OD was 0.05. This broth culture was incubated for up to two hours at 200 rpm, 37°C until mid-log phase under aerobic conditions (approximately 3-4 hours) prior to being used for further cell infections under anaerobic conditions.

The minimum amount of bacteria required to provoke a significant response from CFBE41o- cells (0-4h, anaerobic conditions) was determined by screening of 3 different P. aeruginosa isolates as described (Bertelsen, Elborn et al. 2019). Growth curves of P. histicola and P. aeruginosa under anaerobic conditions revealed no differences in the growth rates between the two species (S1 Fig).

Cell culture

The F508del homozygote cystic fibrosis cell line (CFBE41o-) was maintained in antibiotic free minimum essential media (MEM, Gibco), supplemented with 10% heat inactivated foetal bovine serum (FBS, Gibco) and 5% L-Glutamine (Gibco) under standard cell culture conditions (37°C, 5% CO2, 95% mixed gas). All tissue culture flasks and plates were pre-coated with a 1 % PurCol type 1 collagen solution (Nutacon) and passaged as described previously (Buchanan, Ernst et al. 2009). HEK-293-TLR2, HEK-293-TLR4, HE-293-TLR5 and HEK-293-TLR null cells were maintained as per manufacturer’s instructions (InvivoGen). 

Infection assays

CFBE41o- cells were infected with P. histicola or P. aeruginosa at an MOI (Multiplicity of Infection) of 100 for 4 hours. Both bacteria were grown to mid log phase as described previously (Bertelsen, Elborn et al. 2019). An MOI of 100 was defined by plating all inocula on CBA agar and enumerating viable counts the following morning. Liquid cultures were used to inoculate cells for infection experiments and cells were incubated for up to 4 hours under anaerobic conditions as described above. Cells were incubated under anaerobic conditions for the duration of the experiments as described in supplementary data (S1 Text).

Cell viability assays

To confirm that experimental conditions would not negatively affect cell viability, lactate dehydrogenase release (LDH, Abcam), mitochondrial respiration (MTT) and trypan blue exclusion were assessed after exposure to hypoxia and bacteria.

Briefly, bell death was assessed by measuring LDH release from infected cells and non-infected control cells. 10μL of supernatant was used for each assay as per manufacturer’s instructions (Abcam, ab69693). Analyses of mitochondrial activation (measured by MTT (3- [4, 5-dimethyl thiazol-2yl] – 2, 5 diphenyl tetrazolium bromide) conversion to purple formazan (absorbance λ=570nm)) served as a surrogate for cell viability. Trypan Blue (Sigma) was used in the dye exclusion assay. After incubation and loading onto a Neubauer haemocytometer, cells which appeared blue under the microscope were determined as ‘dead’ and cells appearing white were counted as ‘live cells’. Further details of these assays can be found in the supplement. (S2 Fig (A-C)).

TLR Reporter assays

HEK-293-TLR2, HEK-293-TLR4 and HEK-293-TLR5 cells were purchased from InvivoGen and cultured and transfected as per manufacturer’s instructions. Briefly, HEK-293 cells were maintained in high Glucose DMEM with 10% FBS, L-Glutamine and Pen/Strep. 100μg Blasticidin was added to cells after the second passage and cells were maintained in the media thereafter. Cells were transiently transfected with an NF-�B containing reporter construct plasmid (LyoVec and pNifty-Luc™, InvivoGen) and cells were incubated for 24 hours under standard tissue culture conditions to recover from the transfection. Bacterial infection was carried out as described (Bertelsen, Elborn et al. 2019) and cells were incubated under anaerobic conditions for the duration of the experiments.

Cytoplasmic and nuclear fraction extraction for DNA binding ELISA

…

RT-PCR for TLR and Cytokine gene expression 

…

siRNA knockdown assays

…

Statistical analysis

…”

Q3) Growth rate comparison between P. aeruginosa and P. histicola is not standard. Cells should be inoculated at low enough density to monitor lag phase and tracked for long enough to capture log and stationary phases. Minimally, more than 1 doubling should be measured for a true growth rate calculation. It appears that doubling time here is ~4 hours, but at what part of the growth curve are we looking at? Furthermore, the methods here are lacking and the cited reference (Tunney et al 2007 AJRCCM) does not provide any detail germaine to the method at hand, perhaps it is the wrong reference? Regardless, the authors should provide full methodological detail here (and for other sections). For instance, how did the authors so precisely obtain the exact same number of cells reported for each species for the first time point? This result is important to bolster in order to support claims made by the authors (for instance on lines 360-361).

A3) Both strains were grown under different conditions as they require distinct conditions. P.aeruginosa will grow quite well from a start OD 590 of 0.01, through lag, log and lag phases while Prevotella spp. will not grow well with an initial low inoculum , an takes approx. 18 hours to reach exponential growth. Therefore, conditions were optimized to reflect this. The same number of bacteria were added to each cell infection as defined by OD and plating of CFU/mL, where the OD differed between the two species. The paper by Tunney et al. (2007) is quoted as it refers to the culture of Prevotella spp.

The materials and methods section has now been extended in the manuscript and the supplement. 

Q4) The authors use the term "infection" when referring to the experiments described in Fig 1. Do they mean that they are modeling infection of the airway by applying bacteria to cell culture? Or are they specifically referring to infection of their cells by bacteria becoming internalized? Commonly, for infection assays involving bacteria, there is a question of how many bacteria are internalized versus remain extracellular upon treatment. A useful method is to incubate bacteria with cultured cells, allow for internalization, then treat with antibiotic (e.g. gentamicin) to kill extracellular bacteria, and subsequent washing of cells. Assessment of bacterial internalization can then be performed by cfu plating or qPCR. Since the authors do not report doing this, presumably they do not know if bacteria are internalized or not, and if there are differences in the abundance of internalized Ph versus Pa bacteria. TLR5 signaling I believe does not require internalization, therefore perhaps what they are observing is detection of extracellular bacteria. If this is true, and/or if this is what the authors mean (as seems to be indicated by the graphic depiction in Figure 5), I recommend changing the term "infection" to something like "exposure".

A4) Gentamycin protection assays were carried out to ascertain if bacteria were being internalized. Bacteria were screened for susceptibility to Gentamycin and the lowest concentration (20 µg in 2 hours) was employed for internalization.

Briefly, cells were infected with either P. histicola or P. aeruginosa for 1 or 2 hours, supernatants were removed, cells were washed 3 times with pre-warmed, sterile PBS and media was replaced with media containing concentrations of either 20 μg or 50 μg Gentamycin. 

Cells were incubated anaerobically for an additional 2 hours, and supernatants were plated to ensure extra cellular bacteria were no longer viable. Infected epithelial cells were then lysed with 1 % Saponin (5 minutes incubation, with further gentle manual lysis) and cell lysates plated onto CBA and incubated overnight aerobically for P. aeruginosa or anaerobically for P. histicola. The ability of the bacteria to remain viable in the presence of either 0.5 % Triton – X 100 or saponin was assessed, with saponin being deemed to be suitable for these assays. 

Due to the inability to keep the cells for longer than 4 hours under anaerobic conditions it was not possible to add an additional, lower bacteria-static concentration of gentamycin to the infected cells ie. 10 μg and incubate further, however these assays showed that all extracellular bacteria were non-viable after 1 or 2 hours in the presence of either concentration of Gentamycin. In our hands there was no internalization of these bacterial strains by the cells under our experimental conditions. While there is a possibility that, at the higher concentration of Gentamycin, the Gentamycin may have been internalized and killed any intracellular bacteria, the lack of intracellular bacteria detected with either concentrations indicates that these bacteria were not up-taken by the epithelial cells and remained extracellular for the duration of our experiments.

Infection is not only defined by bacterial uptake. Exposure of our bacterial strains to epithelial cells caused an inflammatory response we refer to as infection. Both Pseudomonas and Prevotella strains secrete outer membrane vehicles (OMVs), which are up taken by cells and provoke an inflammatory response (Metruccio, Evans et al. 2016, Yang, Chen et al. 2019). This shows that infection and subsequent inflammation cannot solely be attributed to bacterial uptake.

Figure 5 is a schematic representation of the investigated signalling pathways induced by infections of CFBE41o- cells. The interaction of bacteria with the tested TLRs (TLR5, TLR2) should occur in the first instance at the cell membrane. However, in this study we did not investigate at which time point TLR signalling would cease. 

Q5) P.histicola and P.aeruginosa – there should be a space following “P.” throughout the text

A5) We thank the reviewer for pointing this out. This has been modified throughout the text.

Q6) Lines 90-91 should be cited. In general, more statements throughout the text that are presented as facts, should be supported by references to published literature. Another example is the first line of the Discussion lines 339-340, and lines 367-369.

A6) We thank the reviewer for pointing this out. This has been modified and reads as follows:

Line 90-91: “In CF airway diseases, the canonical NF-�B signalling pathway, consisting of the p65 and p50 subunits, and its role in inflammation in CF has been extensively investigated (Blackwell, Stecenko et al. 2001, Kelly, Williams et al. 2013, Bertelsen, Elborn et al. 2019), ….”

Line 339-340 (now line 371-372): “In CF airways, infections with bacterial species such as P.aeruginosa, Burkholderia cepecia complex, Haemophilus influenza and S.aureus are well investigated (Tang, Turvey et al. 2014, Zemanick and Hoffman 2016, Bevivino, Bacci et al. 2019).”

Line 367-369 (now line 411-413): “TLRs (especiallyTLR2, 4 and 5) in the CF lung are one of the most common innate immune defences activated in response to infection leading to the activation of the classical NF-kB-activation and subsequent pro-inflammatory cytokine release (Greene, Carroll et al. 2005, John, Yildirim et al. 2010, Kelly, Canning et al. 2013, Kosamo, Hisert et al. 2020).”

Q7) Line 454, typo “aerugiosa”, 

 Line 362, typo “.f” at the end of the line.

A7) We thank the reviewer for pointing this out. This has been corrected.

Reviewer #3: 

Q8). The results should be validated in another CF cell line.

A8) The CFBE41o- cell line is homozygous for F508del and has been extremely well characterized in terms of both their immune responses and their ability to polarize and form tight junctions under appropriate cell culture conditions. It further displays defective cAMP dependent chloride transport while maintaining intact calcium dependent chloride transport. For these reasons, this cell line is deemed to be a very robust and appropriate model for CF studies, and we opted to work with this cell line over other, less well-characterized cell lines for these reasons.

Q9) The authors report the mRNA levels of IL-8, IL-6 and CXCL1 (Figure 1). They should also report the corresponding secreted protein levels.

A9) Aerobic experiments showed that CFBE and HBE cells, under submerged conditions did not secrete significant levels of cytokines at 4 hours in response to various concentrations of LPS, P. aeruginosa infection with a range of clinical isolates and the lab strain PAO1 as well as bacterial whole cell lysates. Significant levels of secreted cytokines were only observed at 6 hours post infection/ exposure to the various treatments. As we could only maintain the cells for 4 hours under anaerobic conditions, we opted to use qRT-PCR cytokine gene expression to assess cytokine responses to infection. We were able to confirm that, under aerobic conditions, cells exposed to P. aeruginosa infection displayed similar levels of cytokine gene expression at 2 and 4 hours as compared to those incubated anaerobically and corresponding cytokine release was observed from these cells at 6 hours. This indicates that should we have been able to maintain the cells for longer under anaerobic conditions we would have seen IL-6, IL-8 and CXCL-1 release from the epithelial cells in response to exposure to the bacteria, as indicated by the gene expression.

 

References:

Bertelsen, A., et al. (2019). "Infection with Prevotella nigrescens induces TLR2 signalling and low levels of p65 mediated inflammation in Cystic Fibrosis bronchial epithelial cells." Journal of Cystic Fibrosis.

Bevivino, A., et al. (2019). "Deciphering the Ecology of Cystic Fibrosis Bacterial Communities: Towards Systems-Level Integration." Trends Mol Med 25(12): 1110-1122.

 Despite over a decade of cystic fibrosis (CF) microbiome research, much remains to be learned about the overall composition, metabolic activities, and pathogenicity of the microbes in CF airways, limiting our understanding of the respiratory microbiome's relation to disease. Systems-level integration and modeling of host-microbiome interactions may allow us to better define the relationships between microbiological characteristics, disease status, and treatment response. In this way, modeling could pave the way for microbiome-based development of predictive models, individualized treatment plans, and novel therapeutic approaches, potentially serving as a paradigm for approaching other chronic infections. In this review, we describe the challenges facing this effort and propose research priorities for a systems biology approach to CF lung disease.

Blackwell, T. S., et al. (2001). "Dysregulated NF-kappaB activation in cystic fibrosis: evidence for a primary inflammatory disorder." Am J Physiol Lung Cell Mol Physiol 281(1): L69-70.

Buchanan, P. J., et al. (2009). "Role of CFTR, Pseudomonas aeruginosa and Toll-like receptors in cystic fibrosis lung inflammation." Biochem Soc Trans 37(Pt 4): 863-867.

 CF (cystic fibrosis) is a severe autosomal recessive disease most common in Northwest European populations. Underlying mutations in the CFTR (CF transmembrane conductance regulator) gene cause deregulation of ion transport and subsequent dehydration of the airway surface liquid, producing a viscous mucus layer on the airway surface of CF patients. This layer is readily colonized by bacteria such as Pseudomonas aeruginosa. Owing to the resulting environment and treatment strategies, the bacteria acquire genetic modifications such as antibiotic resistance, biofilm formation, antimicrobial peptide resistance and pro-inflammatory lipid A structures. Lipid A is a component of the lipopolysaccharide cell wall present on bacteria and is recognized by TLR4 (Toll-like receptor 4). Its detection elicits a pro-inflammatory response that is heightened over time due to the addition of fatty acids to the lipid A structure. Eradication of bacteria from the lungs of CF patients becomes increasingly difficult and eventually leads to mortality. In the present review, we describe the role of lipid A as a virulent factor of Ps. aeruginosa; however, it appears that further work is needed to investigate the role of CFTR in the innate immune response and in modifying the pathogen-host interaction.

Gilpin, D. F., et al. (2017). "Evidence of persistence of Prevotella spp. in the cystic fibrosis lung." J Med Microbiol: 825-832.

 PURPOSE: Prevotella spp. represent a diverse genus of bacteria, frequently identified by both culture and molecular methods in the lungs of patients with chronic respiratory infection. However, their role in the pathogenesis of chronic lung infection is unclear; therefore, a more complete understanding of their molecular epidemiology is required. METHODOLOGY: Pulsed Field Gel Electrophoresis (PFGE) and Random Amplified Polymorphic DNA (RAPD) assays were developed and used to determine the degree of similarity between sequential isolates (n=42) from cystic fibrosis (CF) patients during periods of clinical stability and exacerbation. RESULTS: A wide diversity of PFGE and RAPD banding patterns were observed, demonstrating considerable within-genus heterogeneity. In 8/12 (66.7 %) cases, where the same species was identified at sequential time points, pre- and post-antibiotic treatment of an exacerbation, PFGE/RAPD profiles were highly similar or identical. Congruence was observed between PFGE and RAPD (adjusted Rand coefficient, 0.200; adjusted Wallace RAPD->PFGE 0.459, PFGE->RAPD 0.128). Furthermore, some isolates could not be adequately assigned a species name on the basis of 16S rRNA analysis: these isolates had identical PFGE/RAPD profiles to Prevotellahisticola. CONCLUSION: The similarity in PFGE and RAPD banding patterns observed in sequential CF Prevotella isolates may be indicative of the persistence of this genus in the CF lung. Further work is required to determine the clinical significance of this finding, and to more accurately distinguish differences in pathogenicity between species.

Greene, C. M., et al. (2005). "TLR-induced inflammation in cystic fibrosis and non-cystic fibrosis airway epithelial cells." J Immunol 174(3): 1638-1646.

 Cystic fibrosis (CF) is a genetic disease characterized by severe neutrophil-dominated airway inflammation. An important cause of inflammation in CF is Pseudomonas aeruginosa infection. We have evaluated the importance of a number of P. aeruginosa components, namely lipopeptides, LPS, and unmethylated CpG DNA, as proinflammatory stimuli in CF by characterizing the expression and functional activity of their cognate receptors, TLR2/6 or TLR2/1, TLR4, and TLR9, respectively, in a human tracheal epithelial line, CFTE29o(-), which is homozygous for the DeltaF508 CF transmembrane conductance regulator mutation. We also characterized TLR expression and function in a non-CF airway epithelial cell line 16HBE14o(-). Using RT-PCR, we demonstrated TLR mRNA expression. TLR cell surface expression was assessed by fluorescence microscopy. Lipopeptides, LPS, and unmethylated CpG DNA induced IL-8 and IL-6 protein production in a time- and dose-dependent manner. The CF and non-CF cell lines were largely similar in their TLR expression and relative TLR responses. ICAM-1 expression was also up-regulated in CFTE29o(-) cells following stimulation with each agonist. CF bronchoalveolar lavage fluid, which contains LPS, bacterial DNA, and neutrophil elastase (a neutrophil-derived protease that can activate TLR4), up-regulated an NF-kappaB-linked reporter gene and increased IL-8 protein production in CFTE29o(-) cells. This effect was abrogated by expression of dominant-negative versions of MyD88 or Mal, key signal transducers for TLRs, thereby implicating them as potential anti-inflammatory agents for CF.

John, G., et al. (2010). "TLR-4-mediated innate immunity is reduced in cystic fibrosis airway cells." Am J Respir Cell Mol Biol 42(4): 424-431.

 Airway epithelial cells contribute to the inflammatory response of the lung, and their innate immune response is primarily mediated via Toll-like receptor (TLR) signaling. Cystic fibrosis (CF) airways are chronically infected with Pseudomonas aeruginosa, suggesting a modified immune response in CF. We investigated the TLR-4 expression and the inflammatory profile (IL-8 and IL-6 secretion) in CF bronchial epithelial cell line CFBE41o- and its CF transmembrane ion condcutance regulator (CFTR)-corrected counterpart grown under air-liquid interface conditions after stimulation with lipopolysaccharide (LPS) from gram-negative bacteria. In CFTR-corrected cells, IL-8 and IL-6 secretions were constitutively activated but significantly increased after LPS stimulation compared with CFBE41o-. Blocking TLR-4 by a specific antibody significantly inhibited IL-8 secretion only in CFTR-corrected cells. Transfection with specific siRNA directed against TLR-4 mRNA significantly reduced the response to LPS in both cell lines. Fluorescence-activated cell sorter analysis revealed significantly higher levels of TLR-4 surface expression in CFTR-corrected cells. In histologic lung sections of patients with CF, the TLR-4 expression in the bronchial epithelium was significantly reduced compared with healthy control subjects. In CF the loss of CFTR function appears to decrease innate immune responses, possibly by altering the expression of TLR-4 on airway epithelial cells. This may contribute to chronic bacterial infection of CF airways.

Kelly, C., et al. (2013). "Toll-like receptor 4 is not targeted to the lysosome in cystic fibrosis airway epithelial cells." Am J Physiol Lung Cell Mol Physiol 304(5): L371-382.

 The innate immune response to bacterial infection is mediated through Toll-like receptors (TLRs), which trigger tightly regulated signaling cascades through transcription factors including NF-kappaB. LPS activation of TLR4 triggers internalization of the receptor-ligand complex which is directed toward lysosomal degradation or endocytic recycling. Cystic fibrosis (CF) patients display a robust and uncontrolled inflammatory response to bacterial infection, suggesting a defect in regulation. This study examined the intracellular trafficking of TLR4 in CF and non-CF airway epithelial cells following stimulation with LPS. We employed cells lines [16hBE14o-, CFBE41o- (CF), and CFTR-complemented CFBE41o-] and confirmed selected experiments in primary nasal epithelial cells from non-CF controls and CF patients (F508del homozygous). In control cells, TLR4 expression (surface and cytoplasmic) was reduced after LPS stimulation but remained unchanged in CF cells and was accompanied by a heightened inflammatory response 24 h after stimulation. All cells expressed markers of the early (EEA1) and late (Rab7b) endosomes at basal levels. However, only CF cells displayed persistent expression of Rab7b following LPS stimulation. Rab7 variants may directly internalize bacteria to the Golgi for recycling or to the lysosome for degradation. TLR4 colocalized with the lysosomal marker LAMP1 in 16 hBE14o- cells, suggesting that TLR4 is targeted for lysosomal degradation in these cells. However, this colocalization was not observed in CFBE41o- cells, where persistent expression of Rab7 and release of proinflammatory cytokines was detected. Consistent with the apparent inability of CF cells to target TLR4 toward the lysosome for degradation, we observed persistent surface and cytoplasmic expression of this pathogen recognition receptor. This defect may account for the prolonged cycle of chronic inflammation associated with CF.

Kelly, C., et al. (2013). "Expression of the nuclear factor-kappaB inhibitor A20 is altered in the cystic fibrosis epithelium." Eur Respir J 41(6): 1315-1323.

 A20 is a lipopolysaccharide (LPS)-inducible, cytoplasmic zinc finger protein, which inhibits Toll-like receptor-activated nuclear factor (NF)-kappaB signalling by deubiquitinating tumour necrosis factor receptor-associated factor (TRAF)-6. The action of A20 is facilitated by complex formation with ring finger protein (RNF)-11, Itch and TAX-1 binding protein-1 (TAX1BP1). This study investigated whether the expression of A20 is altered in the chronically inflamed cystic fibrosis (CF) airway epithelium. Nasal epithelial cells from CF patients (F508del homozygous), non-CF controls and immortalised epithelial cells (16HBE14o- and CFBE41o-) were stimulated with LPS. Cytoplasmic expression of A20 and expression of NF-kappaB subunits were analysed. Formation of the A20 ubiquitin editing complex was also investigated. In CFBE41o-, peak LPS-induced A20 expression was delayed compared with 16HBE14o- and fell significantly below basal levels 12-24 h after LPS stimulation. This was confirmed in primary CF airway cells. Additionally, a significant inverse relationship between A20 and p65 expression was observed. Inhibitor studies showed that A20 does not undergo proteasomal degradation in CFBE41o-. A20 interacted with TAX1BP1, RNF11 and TRAF6 in 16HBE14o- cells, but these interactions were not observed in CFBE41o-. The expression of A20 is significantly altered in CF, and important interactions with complex members and target proteins are lost, which may contribute to the state of chronic NF-kappaB-driven inflammation.

Kosamo, S., et al. (2020). "Strong toll-like receptor responses in cystic fibrosis patients are associated with higher lung function." J Cyst Fibros 19(4): 608-613.

 BACKGROUND: Cystic fibrosis (CF) airways disease varies widely among patients with identical cystic fibrosis transmembrane conductance regulator (CFTR) genotypes. Robust airway inflammation is thought to be deleterious in CF; inter-individual variation in Toll-like receptor (TLR)-mediated innate immune inflammatory responses (TMIIR) might account for a portion of the phenotypic variation. We tested if TMIIR in people with CF are different than those of healthy controls, and whether higher TMIIR in people with CF are associated with reduced lung function. METHODS: We cultured whole blood from clinically stable subjects with CF (n = 76) and healthy controls (n = 45) with TLR agonists, and measured cytokine production and expression of TLR-associated genes. We tested for differences in TLR-stimulated cytokine levels between subjects with CF and healthy subjects, and for associations between cytokine and gene expression levels with baseline lung function (forced expiratory volume in one second percent predicted (FEV1%)) and decline in FEV1% over time. RESULTS: TMIIR in blood from subjects with CF were lower than in healthy controls. Expression of TLR regulators SARM1, TOLLIP, and AKT1 were downregulated in CF. In subjects with CF we found that lower TLR4-agonist-induced IL-8 was associated with lower FEV1% at enrollment (p<0.001) and with greater five year FEV1% decline (p<0.001). CONCLUSIONS: TMIIR were lower in people with CF relative to healthy controls; however, unexpectedly, greater whole blood TMIIR were positively associated with lung function in people with CF. These findings suggest a complex interaction between inflammation and disease in people with CF.

Metruccio, M. M., et al. (2016). "Pseudomonas aeruginosa Outer Membrane Vesicles Triggered by Human Mucosal Fluid and Lysozyme Can Prime Host Tissue Surfaces for Bacterial Adhesion." Front Microbiol 7: 871.

 Pseudomonas aeruginosa is a leading cause of human morbidity and mortality that often targets epithelial surfaces. Host immunocompromise, or the presence of indwelling medical devices, including contact lenses, can predispose to infection. While medical devices are known to accumulate bacterial biofilms, it is not well understood why resistant epithelial surfaces become susceptible to P. aeruginosa. Many bacteria, including P. aeruginosa, release outer membrane vesicles (OMVs) in response to stress that can fuse with host cells to alter their function. Here, we tested the hypothesis that mucosal fluid can trigger OMV release to compromise an epithelial barrier. This was tested using tear fluid and corneal epithelial cells in vitro and in vivo. After 1 h both human tear fluid, and the tear component lysozyme, greatly enhanced OMV release from P. aeruginosa strain PAO1 compared to phosphate buffered saline (PBS) controls ( approximately 100-fold). Transmission electron microscopy (TEM) and SDS-PAGE showed tear fluid and lysozyme-induced OMVs were similar in size and protein composition, but differed from biofilm-harvested OMVs, the latter smaller with fewer proteins. Lysozyme-induced OMVs were cytotoxic to human corneal epithelial cells in vitro and murine corneal epithelium in vivo. OMV exposure in vivo enhanced Ly6G/C expression at the corneal surface, suggesting myeloid cell recruitment, and primed the cornea for bacterial adhesion ( approximately 4-fold, P < 0.01). Sonication disrupted OMVs retained cytotoxic activity, but did not promote adhesion, suggesting the latter required OMV-mediated events beyond cell killing. These data suggest that mucosal fluid induced P. aeruginosa OMVs could contribute to loss of epithelial barrier function during medical device-related infections.

Muhlebach, M. S., et al. (2018). "Anaerobic bacteria cultured from cystic fibrosis airways correlate to milder disease: a multisite study." Eur Respir J 52(1).

 Anaerobic and aerobic bacteria were quantitated in respiratory samples across three cystic fibrosis (CF) centres using extended culture methods. Subjects aged 1-69 years who were clinically stable provided sputum (n=200) or bronchoalveolar lavage (n=55). 18 anaerobic and 39 aerobic genera were cultured from 59% and 95% of samples, respectively; 16 out of 57 genera had a >/=5% prevalence across centres.Analyses of microbial communities using co-occurrence networks in sputum samples showed groupings of oral, including anaerobic, bacteria, whereas typical CF pathogens formed distinct entities. Pseudomonas was associated with worse nutrition and F508del genotype, whereas anaerobe prevalence was positively associated with pancreatic sufficiency, better nutrition and better lung function. A higher total anaerobe/total aerobe CFU ratio was associated with pancreatic sufficiency and better nutrition. Subjects grouped by factor analysis who had relative dominance of anaerobes over aerobes had milder disease compared with a Pseudomonas-dominated group with similar proportions of subjects that were homozygous for F508del.In summary, anaerobic bacteria occurred at an early age. In sputum-producing subjects anaerobic bacteria were associated with milder disease, suggesting that targeted eradication of anaerobes may not be warranted in sputum-producing CF subjects.

O'Neill, K., et al. (2015). "Reduced bacterial colony count of anaerobic bacteria is associated with a worsening in lung clearance index and inflammation in cystic fibrosis." PLoS One 10(5): e0126980.

 Anaerobic bacteria have been identified in abundance in the airways of cystic fibrosis (CF) subjects. The impact their presence and abundance has on lung function and inflammation is unclear. The aim of this study was to investigate the relationship between the colony count of aerobic and anaerobic bacteria, lung clearance index (LCI), spirometry and C-Reactive Protein (CRP) in patients with CF. Sputum and blood were collected from CF patients at a single cross-sectional visit when clinically stable. Community composition and bacterial colony counts were analysed using extended aerobic and anaerobic culture. Patients completed spirometry and a multiple breath washout (MBW) test to obtain LCI. An inverse correlation between colony count of aerobic bacteria (n = 41, r = -0.35; p = 0.02), anaerobic bacteria (n = 41, r = -0.44, p = 0.004) and LCI was observed. There was an inverse correlation between colony count of anaerobic bacteria and CRP (n = 25, r = -0.44, p = 0.03) only. The results of this study demonstrate that a lower colony count of aerobic and anaerobic bacteria correlated with a worse LCI. A lower colony count of anaerobic bacteria also correlated with higher CRP levels. These results indicate that lower abundance of aerobic and anaerobic bacteria may reflect microbiota disruption and disease progression in the CF lung.

Tang, A. C., et al. (2014). "Current concepts: host-pathogen interactions in cystic fibrosis airways disease." Eur Respir Rev 23(133): 320-332.

 Chronic infection and inflammation are defining characteristics of cystic fibrosis (CF) airway disease. Conditions within the airways of patients living with CF are conducive to colonisation by a variety of opportunistic bacterial, viral and fungal pathogens. Improved molecular identification of microorganisms has begun to emphasise the polymicrobial nature of infections in the CF airway microenvironment. Changes to CF airway physiology through loss of cystic fibrosis transmembrane conductance regulator functionality result in a wide range of immune dysfunctions, which permit pathogen colonisation and persistence. This review will summarise the current understanding of how CF pathogens infect, interact with and evade the CF host.

Tunney, M. M., et al. (2008). "Detection of anaerobic bacteria in high numbers in sputum from patients with cystic fibrosis." Am J Respir Crit Care Med 177(9): 995-1001.

 RATIONALE: Pulmonary infection in cystic fibrosis (CF) is polymicrobial and it is possible that anaerobic bacteria, not detected by routine aerobic culture methods, reside within infected anaerobic airway mucus. OBJECTIVES: To determine whether anaerobic bacteria are present in the sputum of patients with CF. METHODS: Sputum samples were collected from clinically stable adults with CF and bronchoalveolar lavage fluid (BALF) samples from children with CF. Induced sputum samples were collected from healthy volunteers who did not have CF. All samples were processed using anaerobic bacteriologic techniques and bacteria within the samples were quantified and identified. MEASUREMENTS AND MAIN RESULTS: Anaerobic species primarily within the genera Prevotella, Veillonella, Propionibacterium, and Actinomyces were isolated in high numbers from 42 of 66 (64%) sputum samples from adult patients with CF. Colonization with Pseudomonas aeruginosa significantly increased the likelihood that anaerobic bacteria would be present in the sputum. Similar anaerobic species were identified in BALF from pediatric patients with CF. Although anaerobes were detected in induced sputum samples from 16 of 20 volunteers, they were present in much lower numbers and were generally different species compared with those detected in CF sputum. Species-dependent differences in the susceptibility of the anaerobes to antibiotics with known activity against anaerobes were apparent with all isolates susceptible to meropenem. CONCLUSIONS: A range of anaerobic species are present in large numbers in the lungs of patients with CF. If these anaerobic bacteria are contributing significantly to infection and inflammation in the CF lung, informed alterations to antibiotic treatment to target anaerobes, in addition to the primary infecting pathogens, may improve management.

Yang, D., et al. (2019). "Dysregulated Lung Commensal Bacteria Drive Interleukin-17B Production to Promote Pulmonary Fibrosis through Their Outer Membrane Vesicles." Immunity 50(3): 692-706 e697.

 Idiopathic pulmonary fibrosis (IPF) is a severe form of lung fibrosis with a high mortality rate. However, the etiology of IPF remains unknown. Here, we report that alterations in lung microbiota critically promote pulmonary fibrosis pathogenesis. We found that lung microbiota was dysregulated, and the dysregulated microbiota in turn induced production of interleukin-17B (IL-17B) during bleomycin-induced mouse lung fibrosis. Either lung-microbiota depletion or IL-17B deficiency ameliorated the disease progression. IL-17B cooperated with tumor necrosis factor-alpha to induce expression of neutrophil-recruiting genes and T helper 17 (Th17)-cell-promoting genes. Three pulmonary commensal microbes, which belong to the genera Bacteroides and Prevotella, were identified to promote fibrotic pathogenesis through IL-17R signaling. We further defined that the outer membrane vesicles (OMVs) that were derived from the identified commensal microbes induced IL-17B production through Toll-like receptor-Myd88 adaptor signaling. Together our data demonstrate that specific pulmonary symbiotic commensals can promote lung fibrosis by regulating a profibrotic inflammatory cytokine network.

Zemanick, E. T., et al. (2013). "Inflammation and airway microbiota during cystic fibrosis pulmonary exacerbations." PLoS One 8(4): e62917.

 BACKGROUND: Pulmonary exacerbations (PEx), frequently associated with airway infection and inflammation, are the leading cause of morbidity in cystic fibrosis (CF). Molecular microbiologic approaches detect complex microbiota from CF airway samples taken during PEx. The relationship between airway microbiota, inflammation, and lung function during CF PEx is not well understood. OBJECTIVE: To determine the relationships between airway microbiota, inflammation, and lung function in CF subjects treated for PEx. METHODS: Expectorated sputum and blood were collected and lung function testing performed in CF subjects during early (0-3d.) and late treatment (>7d.) for PEx. Sputum was analyzed by culture, pyrosequencing of 16S rRNA amplicons, and quantitative PCR for total and specific bacteria. Sputum IL-8 and neutrophil elastase (NE); and circulating C-reactive protein (CRP) were measured. RESULTS: Thirty-seven sputum samples were collected from 21 CF subjects. At early treatment, lower diversity was associated with high relative abundance (RA) of Pseudomonas (r = -0.67, p<0.001), decreased FEV(1%) predicted (r = 0.49, p = 0.03) and increased CRP (r = -0.58, p = 0.01). In contrast to Pseudomonas, obligate and facultative anaerobic genera were associated with less inflammation and higher FEV(1). With treatment, Pseudomonas RA and P. aeruginosa by qPCR decreased while anaerobic genera showed marked variability in response. Change in RA of Prevotella was associated with more variability in FEV(1) response to treatment than Pseudomonas or Staphylococcus. CONCLUSIONS: Anaerobes identified from sputum by sequencing are associated with less inflammation and higher lung function compared to Pseudomonas at early exacerbation. CF PEx treatment results in variable changes of anaerobic genera suggesting the need for larger studies particularly of patients without traditional CF pathogens.

Zemanick, E. T. and L. R. Hoffman (2016). "Cystic Fibrosis: Microbiology and Host Response." Pediatr Clin North Am 63(4): 617-636.

 The earliest descriptions of lung disease in people with cystic fibrosis (CF) showed the involvement of 3 interacting pathophysiologic elements in CF airways: mucus obstruction, inflammation, and infection. Over the past 7 decades, our understanding of CF respiratory microbiology and inflammation has evolved with the introduction of new treatments, increased longevity, and increasingly sophisticated laboratory techniques. This article reviews the current understanding of infection and inflammation and their roles in CF lung disease. It also discusses how this constantly evolving information is used to inform current therapeutic strategies, measures and predictors of disease severity, and research priorities.

---

## [Editor Report · Decision Letter 1]

18 Sep 2020

Toll like Receptor signalling by Prevotella histicola activates alternative NF-κB signalling in Cystic Fibrosis bronchial epithelial cells compared to P. aeruginosa

PONE-D-20-19173R1

Dear Dr. Bettina C Schock,

We’re pleased to inform you that your manuscript has been judged scientifically suitable for publication and will be formally accepted for publication once it meets all outstanding technical requirements.

Kind regards,

Abdelwahab Omri, Pharm B, Ph.D

Academic Editor

PLOS ONE

---

## [Editor Report · Acceptance letter]

29 Sep 2020

PONE-D-20-19173R1 

Toll like Receptor signalling by *Prevotella histicola* activates alternative NF-κB signalling in Cystic Fibrosis bronchial epithelial cells compared to *P. aeruginosa*

Dear Dr. Schock:

I'm pleased to inform you that your manuscript has been deemed suitable for publication in PLOS ONE. Congratulations! Your manuscript is now with our production department. 

Kind regards, 

on behalf of

Dr. Abdelwahab Omri 

Academic Editor

PLOS ONE